# A genetically encoded tool for reconstituting synthetic modulatory neurotransmission and reconnect neural circuits in vivo

Josh D. Hawk [1,2✉], Elias M. Wisdom [2], Titas Sengupta[2], Zane D. Kashlan[2] & Daniel A. Colón-Ramos [2,3✉]

Chemogenetic and optogenetic tools have transformed the field of neuroscience by facilitating the examination and manipulation of existing circuits. Yet, the field lacks tools that enable rational rewiring of circuits via the creation or modification of synaptic relationships. Here we report the development of HySyn, a system designed to reconnect neural circuits in vivo by reconstituting synthetic modulatory neurotransmission. We demonstrate that genetically targeted expression of the two HySyn components, a *Hydra*-derived neuropeptide and its receptor, creates de novo neuromodulatory transmission in a mammalian neuronal tissue culture model and functionally rewires a behavioral circuit in vivo in the nematode *Caenorhabditis elegans*. HySyn can interface with existing optogenetic, chemogenetic and pharmacological approaches to functionally probe synaptic transmission, dissect neuropeptide signaling, or achieve targeted modulation of specific neural circuits and behaviors.

[1] Grass Laboratory, Marine Biological Laboratory, Woods Hole, MA, USA. [2] Department of Neuroscience and Department of Cell Biology, Program in Cellular Neuroscience, Neurodegeneration and Repair, Yale University School of Medicine, New Haven, CT, USA. [3] Instituto de Neurobiología, Recinto de Ciencias Médicas, Universidad de Puerto Rico, San Juan, Puerto Rico. ✉email: jsh.hawk@gmail.com; daniel.colon-ramos@yale.edu

Recent advances in optogenetic and chemogenetic tools have provided unprecedented in vivo access to neurons, enabling manipulation of neuronal activity patterns at will[1,2] and facilitating examination of their roles in behaviors. Moreover, synapse-specific labeling methods, such as Trans-TANGO[3] and synaptic GRASP[4], have enabled visualization of individual synaptic connections in the context of the intact circuits, and in the case of Trans-TANGO, genetic access to transsynaptically labeled neurons. Tools like these have transformed the field of neuroscience by facilitating the examination and manipulation of existing circuits. Yet, the field lacks tools that enable rational rewiring of circuits via the creation or modification of synaptic relationships. Engineering de novo circuit relationships could reveal important components of the underlying circuit structures, and rational rewiring of circuits could prove a powerful strategy towards understanding how neuronal relationships generate behaviors.

Ideally, de novo synapses, like other orthogonal genetic tools used for neuronal manipulation, should be generated from specific, controllable modules that enable modification of circuits, but avoid undesirable or uncontrollable interactions with endogenous components of the system. Cross-species transplantation of channels (like Channelrhodopsin) and gap junctions (like the use of Connexins in invertebrates[5–7]) have been powerful in providing the desired modularity and specificity to the engineered systems[2,5,6]. For example, heterologous expression of the *Drosophila* allostatin receptor in vertebrates can be used to silence neurons upon the exogenous application of the allostatin peptide[4]. Similarly, expression in *C. elegans* neurons of the *Drosophila* HisCl channel enables chemogenetic inhibition of neuronal activity[8]. In invertebrates, heterologous expression of vertebrate gap junction proteins, called connexins, within specific neurons of the *C. elegans* nerve ring result in the predictable creation of artificial electrical connections between adjacent neurons[5,6], the rewiring of the circuit, and the recoding of a learned behavioral preference[7]. Yet, no existing technology enables the purposeful creation of new, synthetic and manipulatable chemical synaptic connections for re-configuring neural networks in diverse organisms.

For these reasons, we developed a system that allows the engineering of synthetic relationships between neurons through the targeted reconstitution of modulatory neurotransmission between selected partners, eliciting orthogonal circuit control over neuromodulatory connectivity. Because neuromodulation is a powerful way of re-configuring neural circuits to produce distinct behavioral outcomes, because neuromodulation is not constrained by the architecture of the nervous system, and because the role of neuromodulators and their long-range effects remains largely unexplored[9,10], we focused our efforts on a system that enables reconstitution of synthetic modulatory chemical synaptic connectivity. We note that we use the term "chemical synaptic connectivity" to refer to peptidergic synaptic relationships between neurons, be it junctional relationships between abutting partners, or partners that communicate via volumetric neurotransmission and at a distance. We prioritized the design of a system that would be (1) versatile to function in a wide range of cell types and organisms, (2) modular to allow independent genetic targeting of pre- and postsynaptic components, (3) specific to modulate only the intended target cells while being inert to endogenous neurotransmission, (4) robust by targeting conserved intracellular signaling cascades, and (5) synergistic to interface with existing optogenetic and chemogenetic technologies. Informed by these goals, we engineered "HySyn", a *Hydra*-derived, two-component system that creates synthetic neuromodulatory connections to manipulate intracellular calcium within in vivo neural circuits (Fig. 1a, left).

## Results

In *Hydra*, the loosely connected nerve net uses neuropeptides to produce volumetric neurotransmission[11]. We reasoned that this property of *Hydra* peptidergic synapses, if reconstituted in other systems, would enable examination of adjacent chemical synaptic relationships, as well as the reconstitution of new functional connections between neurons that are not adjacent to each other. It would also allow orthogonal neuromodulation of targeted endogenous circuits at a distance, similar to how neuromodulatory systems work in vivo. We, therefore, built HySyn using a *Hydra*-derived RFamide-related peptide (HyRFamide I/II), and its postsynaptic cognate receptor (HyNaC 2/7/9) that fluxes calcium[12,13]. Importantly, the primary sequence of this *Hydra*-derived peptide is distinct from RFamide-related peptides found in other organisms[14,15] and specific to its receptor[12]. Because presynaptic neuropeptide processing, transport and release mechanisms[14] and postsynaptic calcium signaling are conserved throughout evolution[16–18], expression of this neuropeptide ligand-receptor pair, while orthogonal, would still harness fundamental and conserved signaling mechanisms in the desired cells. The harnessing of conserved biological pathways with an orthogonal and specific neuropeptide–receptor pair achieved two goals: (1) it allowed reconstitution of neuronal connections with minimal components and (2) it resulted in a versatile tool for use in different biological contexts. The divergent evolution of this neuropeptide–receptor pair, in the context of conserved cell biology, could be exploited to produce a bipartite synthetic connection whose components would be (individually) inert when expressed in systems other than *Hydra*. Yet when targeted, specific bipartite reconstitution of HySyn could be used to reconfigure modulatory and functional relationships. To emphasize how the HySyn system compares to existing tools: canonical neuroscience approaches, like optogenetics, control specific neurons, whereas the synthetic synaptic approach of HySyn was designed to control relationships between neurons.

To drive heterologous expression, processing, and transport of the cnidarian neuropeptide, we designed a genetically encoded pre-pro-peptide carrier, "HyPep", that harnesses the universality of the neuropeptide processing pathway (Supplementary Fig. 1 and Fig. 1a, "*Hydra* RFamide"). Previous approaches to label neuropeptides concatenated a reporter onto an existing full-length natural neuropeptide precursor[19,20]. We built upon the knowledge of neuropeptide synthesis from these and other studies[15,19–21] to create HyPep as a synthetic pre-pro-peptide carrier that would enable targeting and processing of heterologous neuropeptides using the endogenous neuropeptide processing pathway (Supplementary Fig. 1a). HyPep consists of a signal peptide that directs trafficking, acidic spacers with enzymatic recognition sites for cleavage of the neuropeptide, and the sequence encoding the heterologous neuropeptide itself. We based our signal peptide on neuropeptide Y (Supplementary Fig. 1b, "Signal Peptide"), a ubiquitously expressed neuropeptide in vertebrates[14]. We designed artificial neuropeptide spacers containing consensus cleavage sites (Supplementary Fig. 1b, red lines) for pre-pro-convertase (PC2), a conserved neuropeptide endopeptidase[15]. Because the cross-species alignment of neuropeptides revealed a strong bias for acidic residues between the dibasic cleavage sites, we created acidic linkers between these cleavage sites, which in turn flanked the heterologous cnidarian neuropeptide we sought to express with the HyPep system (Supplementary Fig. 1b, "*Hydra* RFamide").

Expression of the HyPep synthetic pre-pro-peptide carrier in mammalian Neuro2a cells resulted in localization and transport of a fluorescent reporter to the expected intracellular compartments and vesicular release sites (Supplementary Fig. 1c, see also Fig. 3). This observation is consistent with our hypothesis that the

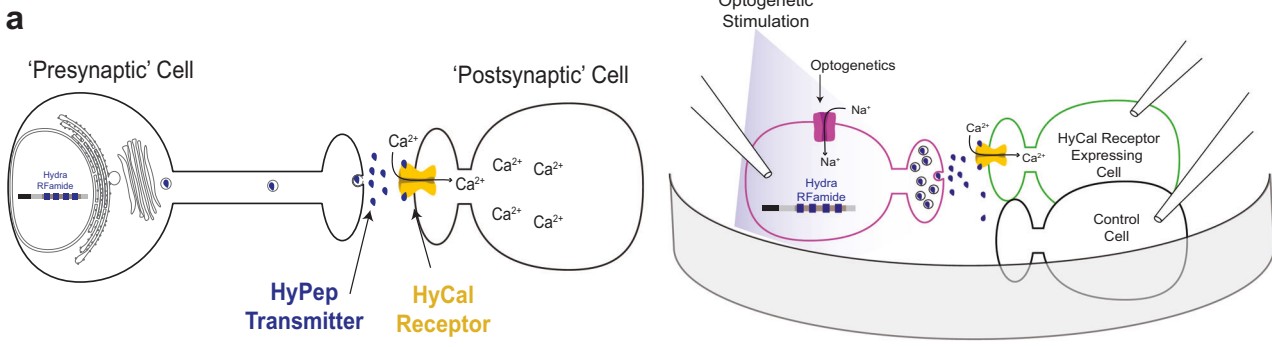

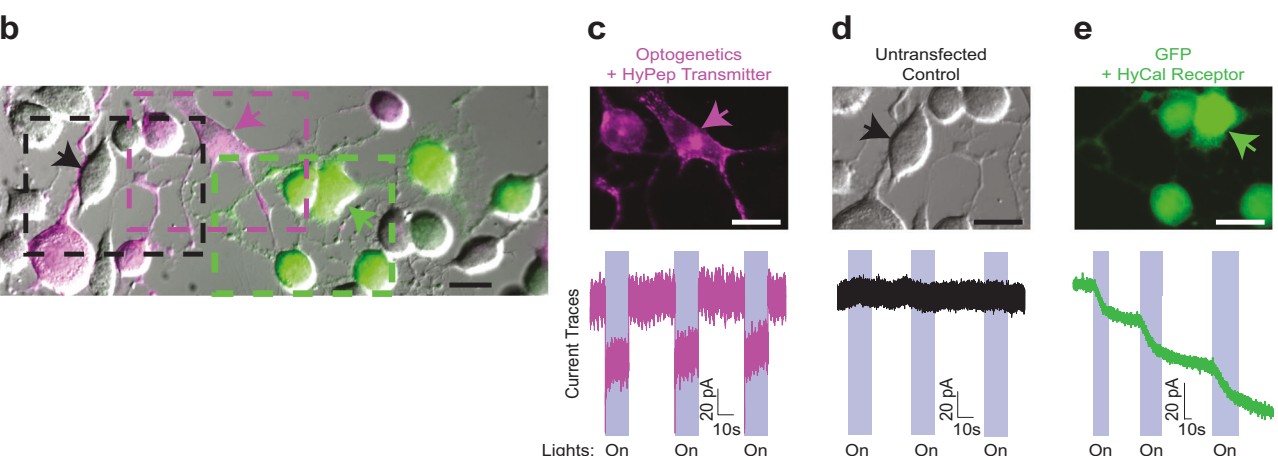

**Fig. 1 Hydra-derived synthetic synapse (HySyn) engineered through heterologous expression of Hydra neuropeptide (HyPep) and receptor (HyCal). a** (left) Schematic describing "HySyn": *Hydra*-derived neuropeptide (blue) is released from the "presynaptic" cell and interacts with the HyCal receptor (yellow) to flux calcium in the "postsynaptic" cell. **a** (right) Schematic illustrating the experimental paradigm to assay HySyn function by whole-cell patch-clamp electrophysiology. **b** Micrograph showing co-culture of Neuro2a neuroblastoma cells expressing either "presynaptic" HyPep with ChRoME (magenta), "postsynaptic" HyCal with GFP (green), or untransfected (grayscale). Boxed regions correspond to the micrographs in **c–e**, with the color of the box matched to the color of the title of the micrograph. **c–e** Identification of cell populations based on fluorescent markers (top, micrograph, 20µm scale bar) enabling whole-cell current recording (bottom, "current traces") from three distinct populations during optogenetic stimulation (480 nm, blue-shaded windows in "current traces", "On"): **c** "presynaptic" cells expressing HyPep with ChRoME (pseudocolored magenta) consistently produced step-wise optogenetic currents, **d** unlabeled cells remain unaltered by optogenetic activation of neighboring HyPep+ cells, and **e** green "postsynaptic" cells expressing the HyCal receptor show an increasing inward current during light stimulation. The downward deflection of traces during optogenetic stimulation in blue windows indicates an inward current, suggesting a depolarizing cation current. The persistence of HyCal current between stimulation (**e**) is consistent with the lack of desensitization of HyCal channel and suggests an accumulation of HyPep neuropeptide in the bath solution. Transfection and reporter expression (micrographs in **b–e**, see also Fig. 2a) was reproducibly observed, including in the 22 cases where successful electrophysiological recordings were made (Supplementary Fig. 2), and in 3 independent populations with GCaMP expression (Fig. 2).

engineered HyPep carrier harnesses the universality of the neuropeptide processing pathway to target the *Hydra* neuropeptide processing, transport, and release. Next, we used whole-cell patch-clamp electrophysiology to test whether the "presynaptic" HyPep, when paired with the "postsynaptic" HyCal receptor, is capable of producing a functional neuromodulatory connection (Fig. 1a). We created an optogenetically excitable population of "presynaptic" cells by co-expressing HyPep with mRuby-labeled ChRoME[22], an optogenetic tool for depolarizing neurons (Fig. 1b, magenta). We then co-cultured these "presynaptic" cells with other cells expressing the "postsynaptic" receptor HyCal (Fig. 1b, green), as well as untransfected control cells (Fig. 1b, unlabeled). As expected[22], optogenetic activation (480 nm) of ChRoME-expressing "presynaptic" cells produced a sustained step-like current consistent with direct channel opening by light (Fig. 1c). Neighboring cells expressing neither the ChRoME optogenetic tool nor the HyCal receptor did not exhibit light-activated

currents (Fig. 1d). But when we optogenetically stimulated the "presynaptic" Neuro2a cells, we observed distinct "postsynaptic" currents in co-cultured cells expressing the "postsynaptic" HyCal receptor (Fig. 1e). These results indicate the existence of synthetic peptidergic connections between the "presynaptic" HyPep- and "postsynaptic" HyCal-expressing cells. Considering the absence of currents without the receptor (Fig. 1d, see also Supplementary Fig. 2), we conclude that these results show the creation of de novo peptidergic synaptic relationships through the specific reconstitution of the HySyn system.

We observed that repeated optogenetic stimulation of the "presynaptic" HyPep-expressing cells produced an integrating current in the "postsynaptic" HyCal receptor-expressing cells (Fig. 1e, bottom). With each stimulation, the inward current (downward trace deflections) increased to a new plateau level. This phenomenon is consistent with the known neuromodulatory properties of the peptide–receptor pair. Specifically, the HyCal

receptor does not desensitize[13], and neuropeptides act at low concentrations across large volumes[15]. Thus, this integrating current suggests that an increasing fraction of HyCal channels open as neuropeptide release and stimulation persist. The selective presence of these currents in HyCal-expressing "postsynaptic" cells aligns with the expected specificity of this system. These data show that combining "presynaptic" HyPep and "postsynaptic" HyCal creates an artificial coupling of activity in a vertebrate neuronally-derived cell culture model, likely by utilizing volume neurotransmission.

Next, we used calcium imaging to examine the extent and reliability of HySyn neuromodulation in a population of cells. We used our established approach to activate "presynaptic" HyPep-expressing cells through optogenetics, but used the red-shifted variant Chrimson[23] (Fig. 2a, magenta cells) that is compatible with GCaMP imaging. In parallel, we monitored activation of "postsynaptic" cells with the calcium-sensitive fluorophore GCaMP6f[24] (Fig. 2a, green cells). In GCaMP-labeled cells lacking the HyCal receptor, we did not observe light-evoked calcium activity (Fig. 2b), which was consistent with our electrophysiological data (Fig. 1d). These results support that HyPep is inert in mouse Neuro2a cells. In contrast, when "presynaptic" HyPep-positive cells were optogenetically stimulated, co-cultured cells expressing the "postsynaptic" HyCal receptor showed rising GCaMP signals (Fig. 2c, responses are organized based on the response magnitude to highlight the frequency of responses). We classified ~34% (14/44) of quantified cells as clear "responders", based on a change in GCaMP signaling 3× standard deviations (or greater) than signal observed prior to light stimulation. Quantification of this increase in calcium over the course of the 4-min stimulation (Fig. 2d and Supplementary Figure 2) highlights an overall doubling of the calcium signal compared to the initial signal, but some cells experienced changes in the GCaMP signal as high as 6-fold. We did not find a clear spatial pattern of activation with respect to the location of the "presynaptic" cells. This observation is consistent with our electrophysiological data and our expectations for a neuropeptide that functions through volume transmission. "Postsynaptic" calcium responses selectively in HyCal-expressing cells provide further support for the reconstitution of the HySyn peptidergic functional connection, and the specificity of HySyn. Our finding that robust calcium responses occur in a third of Neuro2a cells is consistent with a neuromodulatory function (as compared to the more deterministic relationship often evoked by a classical neurotransmitter). Together, these observations support the idea that HySyn creates neuromodulatory connectivity in a vertebrate neuronally-derived cell culture model. Our methods also demonstrate that HySyn is compatible with existing optogenetic and calcium imaging approaches.

We reasoned that if HySyn acts through volume transmission, optogenetic stimulation of a culture of "presynaptic" HyPep-expressing Neuro2a cells would generate a HyPep+ solution that could act as a chemogenetic stimulator when added to a separate culture of "postsynaptic" HyCal-expressing cells (Fig. 2e). To test this idea, we collected the solution from optogenetically stimulated HyPep-expressing Neuro2a cells (HyPep+ solution) and added it to a separate HyCal-expressing cell culture. Prior to adding media, GCaMP signals were stable in all HyCal-expressing "postsynaptic" Neuro2a cells (Fig. 2f), including the 14 cells (~45% of 31 examined) that ultimately showed responses to at least one application of HyPep+ solution. Application of HyPep+ solution produced an increase in the calcium signal, while subsequent washout and application of untreated solution led to a decline in the calcium signal (quantified in Fig. 2g). When the HyPep+ solution was repeatedly added after washout cycles, and individual responding cells were tracked, we observed that of the responding Neuro2a

cells, the majority (8/14) responded during both of the HyPep+ solution applications, and with a calcium rise greater than 3 standard deviations beyond any changes observed before the application. We note, however, that cells responded differentially to the two applications, and that some cells responded to only a single HyPep+ solution application. These results are consistent with the expected neuromodulatory nature of the HySyn system, and suggest that while HyCal modulates the responsiveness of Neuro2a cells, it is not the sole determinant of activity. Importantly, these results provide further support that HySyn functions through volume transmission and that the HyCal receptor may be used with exogenously applied neuropeptide for pharmacological or chemogenetic manipulations of intracellular calcium.

We then sought to determine whether HySyn could be used in vivo to modulate organismal behavior by expressing HySyn in the nematode *C. elegans*. As a first step, we harnessed the well-characterized and stereotyped distribution of synaptic specializations to examine the subcellular localization of HySyn components in vivo. We focused our characterization of HySyn on the AIB interneurons—a pair of symmetric interneurons with a polarized and distinguishable distribution of pre- and post-synaptic sites in the distal and proximal neurite, respectively[25–27] (Fig. 3a). Expression of the transmembrane dense-core vesicle marker, IDA-1/phogrin::mCherry, resulted in the localization of the transmembrane receptor to presynaptic sites in the distal neurite (Fig. 3b). Co-expression of IDA-1::mCherry and GFP-tagged HyPep resulted in colocalization of both components to presynaptic regions of the neurite (Fig. 3c–e), consistent with HyPep being targeted to presynaptic dense core vesicle-rich regions, as expected (and consistent with our observations in Neuro2a cells, Supplementary Fig. 1c). Moreover, expression of the postsynaptic receptor GLR-1::GFP in AIB resulted in its localization to the postsynaptic region of AIB (Fig. 3g–i). Similarly, expression of the postsynaptic HySyn component, the HyCal receptor was enriched in the postsynaptic proximal neurite (Fig. 3h). Together, our data indicate that expression of the HySyn components in vivo results in their expected subcellular localization and trafficking.

To determine whether HySyn could modulate organismal behavior, we then expressed GFP-tagged HyPep throughout the nervous system and the HyCal receptor either in muscle tissues, pan-neuronally, or in GABAergic inhibitory locomotory interneurons (Fig. 3j–l and Supplementary Fig. 3a). We predicted that activating the HyCal receptor in muscles, all neurons, or in inhibitory neurons, would result in paralysis that would prevent proper worm locomotion. Indeed, the reconstitution of HySyn, through the expression of the HyPep and the HyCal in the specific tissues resulted in severely uncoordinated animals (Supplementary Fig. 3a and Supplementary Movies 1, 2). Importantly, expression of the pan-neuronal HyPep alone, or of HyCal alone in the indicated tissues, did not produce any observable phenotypes or locomotory defects (Fig. 3l and Supplementary Fig. 3a), consistent with these elements being inert in vivo in *C. elegans* when expressed on their own. Transgenic animals with pan-neuronal HyPep and muscle-targeted HyCal (i.e., neuromuscular HySyn) performed very little crawling behavior, as illustrated by the trajectory of worms observed for 30 min on an agar pad (Fig. 3k, right) and by the reduced velocity of travel (Fig. 3l, blue). To better quantify the phenotypes, we used DeepLabCut to train a neural network to identify worm postures and quantify worm locomotion[28,29] (see "Methods" section). We observed that in swimming assays, animals with reconstituted HySyn displayed a reduced head radial velocity (Supplementary Fig. 3a, b), consistent with the uncoordinated phenotypes that we detected during locomotion on solid agar surfaces (Fig. 3k, l). We note that HyCal expression in muscle, with HyPep expression in the

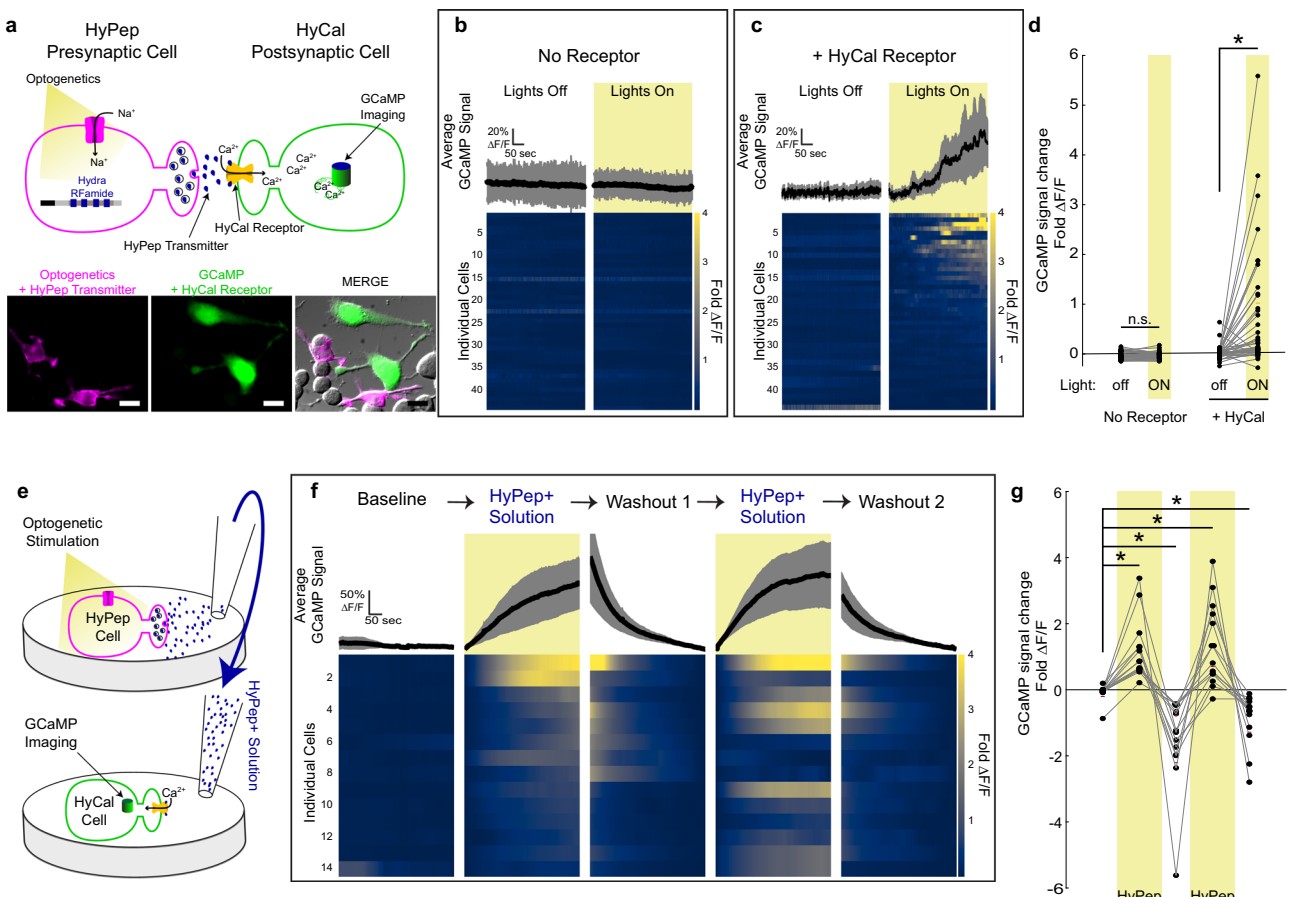

**Fig. 2 HySyn produces volumetric neuromodulation of postsynaptic calcium. a** Schematic illustrating the experimental paradigm used to characterize HyPep to HyCal signaling by using calcium imaging, top. Micrograph illustrating transfected Neuro2a cells used in these experiments, bottom (20 μm scale bar). Chrimson was used for optogenetic stimulation (591 nm, 500 ms at 1 Hz) of "presynaptic" cells expressing the HyPep carrier of the neuropeptide ("*Hydra* RFamide"). GCaMP was expressed in distinct cells either alone (**b**) or with the cognate receptor HyCal (**c**), and both of these two groups of cells were co-cultivated with HyPep "presynaptic" cells. **b** Average GCaMP signals, top bold line with shaded 95% confidence intervals, and individual cell heatmap profiles, bottom, for Neuro2a cells expressing GCaMP alone. Without receptor expression, but even in the presence of HyPep-secreting cells co-expressing the optogenetic tool (Chrimson), GCaMP signals remained stable in these cells both without (left, "Lights Off") and with (right, "Lights On", yellow shading) optogenetic stimulation of the HyPep cells. **c** As in **b**, but in cells also expressing the HyCal receptor in the presence of HyPep-secreting cells co-expressing the optogenetic tool (Chrimson). We observe calcium signal rise over the course of light stimulation. Out of 44 cells, ~34% (14) showed changes in fluorescence over 3 standard deviations beyond the mean change observed prior to light stimulation. **d** Quantification of the change in GCaMP signal (mean $\Delta F/F$ in final 30 s minus mean $\Delta F/F$ in first 30 s) revealed that optogenetic stimulation of co-cultured HyPep-expressing cells co-expressing the optogenetic tool (Chrimson) did not result in stimulation of potential "postsynaptic" cells lacking the HyCal receptor (but expressing GCaMP). In contrast, they resulted in a doubling of intracellular calcium signal in cells expressing the HyCal receptor ($p = 1.0$ for "No Receptor" ($n = 43$) Lights off vs ON, $p = 0.00002$ for "+HyCal Receptor" ($n = 44$) Lights off vs ON). **e** Schematic illustration of solution transfer experiments. After optogenetic stimulation (as in **a** for 5 min), the bathing solution from HyPep-expressing cells ("HyPep+ Solution") was transferred to naive "postsynaptic" cells in another culture dish expressing GCaMP and the HyCal receptor. **f** Following a stable baseline, applying the HyPep+ solution increased the GCaMP signal, and this rise was reversed after washout and applying the fresh bathing solution (Washout). Repeated cycles of washout and application of the HyPep+ solution reproducibly increased the GCaMP signal in the same responding cells ($n = 14$, $p = 0.00004$ 1st addition, $p = 0.0001$ Washout 1, $p = 0.0003$ 2nd addition, $p = 0.0003$ Washout 2). **g** To highlight the kinetics of individual responding cells (from panel **f**), we quantified (as in **d**) GCaMP changes for cells that displayed responses. Error bars (and shaded regions in **b**, **c**, **f**) indicate 95% confidence intervals and * indicates $p < 0.05$ using two-tailed Mann–Whitney–Wilcoxon test with Bonferroni correction for multiple comparisons. Transfection and reporter expression (micrographs in **a**) were reproducibly observed, including in the 22 cases where successful electrophysiological recordings were made (Supplementary Fig. 2), and in 3 independent populations with GCaMP expression (Fig. 2). Source data are provided as a Source Data file.

intestines, was insufficient to drive paralysis or uncoordinated phenotypes (Fig. 3l, "Intestinal HyPep + Muscle HyCal", magenta), consistent with these phenotypes emerging from neuron-to-neuron or neuron-to-muscle synthetic interactions introduced by HySyn reconstitution.

The neuromuscular HySyn establishes a robust and easily observed behavioral phenotype, paralysis. This ease of

observation provides an opportunity for a forward genetic screen to identify novel genes required for neuropeptide processing and release. To test this idea, we examined whether a mutation known to impair neuropeptide function, in the endogenous *C. elegans* pre-pro-convertase enzyme *egl*-3, suppresses the neuromuscular HySyn-induced paralysis phenotype (Supplementary Movie 1 and Fig. 3l). Although the HySyn system components were both

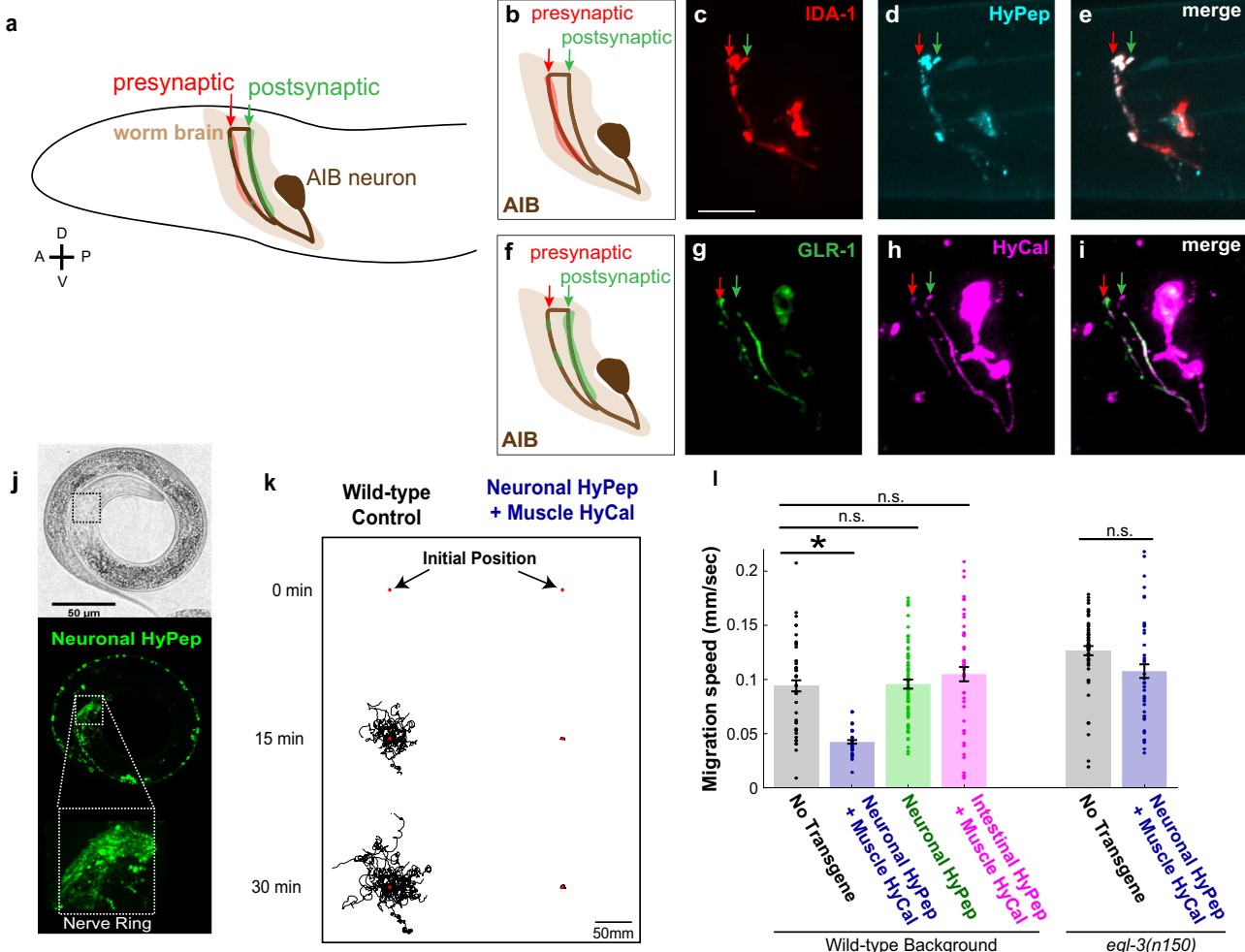

**Fig. 3 HySyn components localize in vivo to appropriate synaptic compartments, and modify animal behavior. a** Schematic of the worm head (black outline) illustrating the polarized distribution of synaptic specializations in the neuron AIB (brown) as determined by previous cell biological and electron microscopy studies[25,26]. Presynaptic sites are restricted to the region of the neurite most distal to the soma (indicated in red, "presynaptic"). Postsynaptic specializations are enriched in the posterior soma-proximal region of the neurite (indicated in green, "postsynaptic"). **b–e** Schematic illustrating enrichment of presynaptic specializations in the neuron AIB (**b** red shading) and representative confocal images of the presynaptic dense-core vesicle marker IDA-1/phogrin::mCherry[46], (**c** red), and the presynaptic component of HySyn, HyPep-eGFP (**d** cyan). Overlay (**e** "merge") showing colocalization of these markers in the presynaptic compartment of the neurite (red arrow marks the position of the distal neurite enriched for presynaptic specializations, as shown in schematic in **b**). The scale bar in **c** is 10 μm. **f–i** Schematic illustrating enrichment of postsynaptic specializations in the neuron AIB (**f** green shading) and representative confocal images of the postsynaptic receptor GLR-1::GFP[26] (**g** green), and the postsynaptic component of HySyn, HyCal-mCherry (**h** magenta). Overlay (**i** merge) showing that within the AIB neurite these two elements colocalize predominantly to the soma-proximal postsynaptic region (green arrow marks the position of the proximal neurite enriched for postsynaptic specializations, as shown in the schematic in **f**). Bright-field (**j** top) and fluorescence (**j** bottom) micrographs show the pattern of expression for HyPep-GFP under the control of a pan-neuronal promoter in the nematode *C. elegans*. The fluorescence pattern (green, bottom panel) shows a dense collection of puncta in the nerve ring (inset), a synaptic-rich neuropil. **k** Illustration of trajectories for 24 worms over a 30 min observation period. Each line represents a single worm track from a commonly aligned initial position (red dot) after either a 0 min (top), 15 min (middle), or 30 min (bottom) monitoring period. Compared to the dispersion of wild-type control (left), transgenic animals expressing the synthetic HySyn connection between neurons and muscles (right, "Neuronal HyPep + Muscle HyCal") showed substantially reduced migration over time. **l** During infrequent bouts of detectable migration in this 30 min interval, those animals expressing the full HySyn system ("Neuronal HyPep + Muscle HyCal", blue) move at a slower speed than control animals without HySyn ("No Transgene", black) (p = 0.000001 for wild-type (n = 43) vs "Neuronal HyPep + Muscle HyCal" (n = 24)). Neither the neuropeptide itself ("Neuronal HyPep", green) nor the receptor in the presence of intestine-produced neuropeptide ("Intestinal HyPep + Muscle HyCal", magenta) altered migration speeds (p = 1.0 for wild-type (n = 43) vs "Neuronal HyPep" (n = 68), p = 1.0 for wild-type (n = 43) vs "Intestinal HyPep + Muscle HyCal" (n = 45)). Migration in animals carrying a mutation of *egl-3(n150)*, a gene required for neuropeptide maturation. The *egl-3(n150)* mutation suppressed the function of the reconstituted HyPep-HyCal connection (right-most bar, blue) based on comparison of *egl-3(n150)* mutants with "Neuronal HyPep + Muscle HyCal" (n = 37) to wild-type (n = 43, p = 1.0) or *egl-3(n150)* mutant animals without transgene expression (n = 52, p = 0.1243). Error bars indicate 95% confidence intervals and * indicates p < 0.05 using two-tailed Mann–Whitney–Wilcoxon test with Bonferroni correction for multiple planned comparisons. Source data are provided as a Source Data file.

present, the ability of HySyn to produce paralysis was suppressed in the *egl-3(n150)* mutants, and worm locomotion reverted to wild-type levels (Fig. 3l, right-most bar). These results support the specificity of this system by demonstrating that functional pre-synaptic HyPep is required to produce the observed paralysis phenotype with postsynaptic HyCal. This dependence on neuropeptide processing also highlights the potential of this system for forward genetic screening to identify novel components of neuropeptide processing and/or release, which can rescue this paralysis.

To then examine whether HySyn could be used to reconnect specific circuits, we used it to repair a broken neuromodulatory connection. In *C. elegans*, the serotonergic neuron called NSM regulates a behavioral switch between two states: a high-speed roaming state that occurs in the absence of food, and a low-speed dwelling state that occurs in the presence of food[30,31]. NSM is an enteric neuron in the pharynx that senses the presence of bacteria (*C. elegans* food) via the use of an acid-sensing ion channel (an ASICs channel) called DEL-7[32]. In the presence of food, NSM releases serotonin to modulate dwelling and, in the absence of food, reduced levels of serotonin result in roaming[30]. Mutants that affect this process, including mutants for DEL-7 (incapable of sensing food), ablations of NSM, or mutants that affect the serotonergic biosynthetic pathway, such as the *tph-1(mg280)* allele[33], all result in animals that abnormally roam even in the presence of food (Fig. 4a, c). The genetics of the serotonergic system in *C. elegans* are well-established, with five distinct serotonin receptors, including two (*ser-4* and *mod-1*) that are expressed in both neurons and muscle to inhibit locomotion[34]. Thus, the modulation of locomotion is genetically and anatomically distributed[30], with parallel pathways that converge from high-level sensory neurons, onto muscle, to mediate slowing. We reasoned that HySyn could be used to bypass interneurons and reconfigure synaptic relationships in vivo to suppress the abnormal roaming behaviors, "repairing" food-induced dwelling in serotonergic mutants. For example, we would expect that HySyn-based functional rewiring between NSM and muscles would suppress the *tph-1* mutants, as HySyn would "repair" the neuromodulatory connection lost in serotonin mutants. But we would not expect HySyn to suppress the *del-7* mutant roaming phenotype, as these animals would be incapable of sensing food, and thus, of releasing HyPep from NSM in the presence of food. Indeed, when we expressed HyPep in NSM, and expressed HyCal in the body wall muscles (pNSM::HySyn, Fig. 4b) of *tph-1* mutant animals, we observed that pNSM::HySyn effectively suppressed roaming, restoring instead a "dwelling-like" state in *tph-1* mutant animals (Fig. 4b, pink). Importantly, pNSM::HySyn animals failed to suppress the *del-7* mutant phenotype, as we predicted, due to their inability to sense food, activate NSM, and release neuropeptides (including HyPep). These findings are consistent with HySyn expression reconstituting an NSM-activity-dependent, neuromodulatory connection. Also consistent with HySyn resulting in the reconstitution of a neuromodulatory connection, we observed we could eliminate the behavioral effects of HySyn in the neuropeptide processing mutant, *egl-3(n150)* (Supplementary Fig. 3c, white). These experiments demonstrate that the HySyn-reconstituted NSM-to-muscle communication is sufficient to reconstitute food-mediated animal dwelling.

Together, these results show that the HySyn system can reconfigure neural circuits in vivo to alter organismal behavior and could be used to probe genetic requirements of neuropeptide signaling and neuromodulation. We note that these types of experiments, which use orthogonal synthetic connections to recover neuromodulatory signaling, are not possible with other existing tools in neuroscience. The approach presented here could be extended to repair connections or create new neuronal

relationships towards understanding circuit functions in varying systems.

Because NSM is selectively active in the presence of food, and because cell-specific expression of HyPep in NSM results in its specific release upon NSM activation, we could examine the extinguishing effects of HySyn in animals in which its release was reduced upon removal from food (and NSM inactivation). We characterized the persistent effects of NSM-released HyPep by monitoring the mean speed in HySyn expressing animals upon their removal from food (Supplementary Fig. 3d, e). We observed that the effect of HySyn (decreased movement, or "dwelling"), in pNSM::HySyn animals was slowly extinguished over a period of roughly 70 min off food, consistent with its role as a long-lived neuromodulator. The half-maximal behavioral effect of HySyn, which was ~40 min after removing animals from food, is consistent with in vivo half-lives of neuropeptides such as vasopressin[35] and neuropeptide Y[36,37]. The duration of neuromodulatory effect of neuropeptides is often limited by non-specific extracellular proteases[38,39], leading us to speculate that this time-dependent reduction in HySyn activity is similarly regulated in vivo.

## Discussion

The versatility of our engineered HySyn system is illustrated by its functioning in both tissue culture Neuro2a cells and in vivo in *C. elegans*. The system is also modular, which facilitates modifying and integrating its components with other approaches. For example, the GFP-labeled version of HyPep can be used on its own to track neuropeptide processing, trafficking or release. Prior work has shown that the HyCal channel, when expressed heterologously, creates a non-inactivating calcium current with direct peptide addition and with dose-dependent currents ranging, from minimal current in the sub-nanomolar range, to maximal current in the micromolar range[12]. Future experiments could examine the ability of synthetic HyPep to induce cellular responses at different concentrations in varying tissues. The HyCal receptor could then be utilized in chemogenetic approaches to pharmacologically enhance calcium in genetically targeted cells. Calcium, in turn, can alter synaptic plasticity, gap junction function, and gene expression at different timescales and based on the persistence of the signal[18]. HyCal activity can be blocked with small-molecule pharmacological agents[12], providing another potential approach for temporal gating of HySyn, an enabling characteristic that should be experimentally validated in varying tissues. Because these components are genetically controlled, they could be linked to conditional approaches based on heterologous inducible promoters[40] or recombination-based activation[41], a form of temporal gating that may be useful in some systems.

The power of the HySyn system lies in the ability to create a new functional connection to bias the relationship between a particular synaptic input and postsynaptic intracellular calcium. We acknowledge, however, that this neuromodulatory aspect also represents a limitation of the system that needs to be further explored in its specific applications. For example, while HySyn provides tissue-level specificity via the controlled expression of its components, we note the persistence of HySyn-mediated effects upon release, both in tissue culture cells and in vivo. The kinetics we observe are consistent with in vivo kinetics observed for other neuropeptides[35–37], yet we anticipate that distinct kinetics of HySyn effects might be achieved in the context of distinct behavioral circuits, and recommend controls, like the ones performed in this study, to characterize the extinction of the HySyn circuit modulation. In this study, we did not examine propagation of action potentials in the context of the neuromodulatory effects of HySyn, another factor that would be useful to characterize in

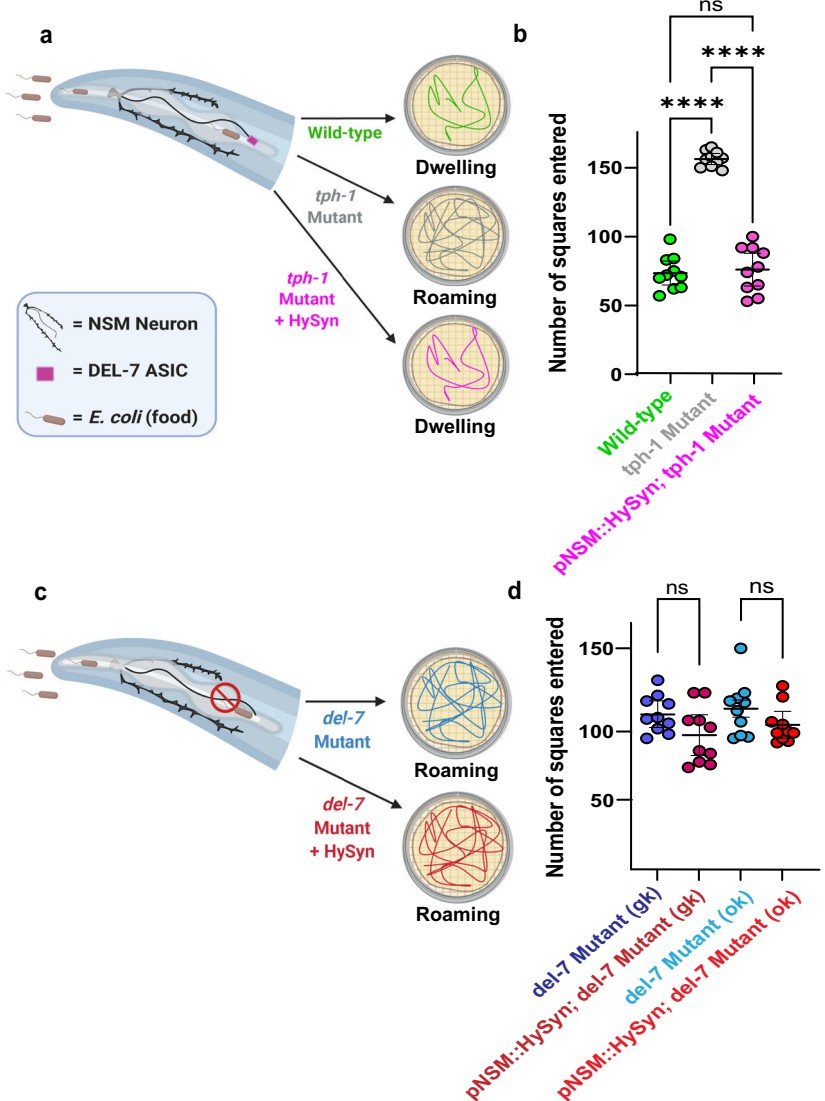

**Fig. 4 Suppression of abnormal behavioral states via targeted reconstitution of HySyn neuromodulatory connections. a** A schematic of a nematode head, and the pharyngeal, food-sensing enteric neuron NSM (black). NSM senses the presence of bacteria (food) via the DEL-7 acid-sensing ion channel (ASIC) receptor (purple box), and releases serotonin, resulting in a "dwelling" behavioral state in wild-type animals (green), which can be scored by quantifying animal displacement in the food-covered Petri dishes, as depicted in the schematic and described[30,32]. Mutant animals for serotonin biosynthesis, such as *tph-1(mg280)*[33], are incapable of releasing serotonin and remain in a state of "roaming" even in the presence of food (gray). Reconstitution of circuit connectivity by using HySyn to connect serotonin-releasing NSM neuron and muscles. Note this configuration of HySyn can suppress the abnormal "roaming" state in *tph-1(mg280)* mutants. **b** Quantification of the roaming and dwelling phenotypes, as described[30,32], for the indicated genotypes. Each dot in the graph represents an individual animal, and a total of ten animals were blindly scored per genotype ($p < 0.0001$ for *tph-1* mutant vs. wild-type, $p < 0.0001$ for *tph-1* mutant vs. pNSM::HySyn; *tph-1* mutant, $n = 10$ independent biological replicates for all groups). **c** As **a**, but for mutant animals lacking the *del-7* receptor, which makes them incapable of sensing food. Note that these animals are wild-type for the serotonin biosynthetic pathway, but phenocopy serotonin mutant animals because the NSM enteric neuron is not activated in the presence of food[30,32]. **d** As **b**, but for two comparable alleles of *del-7* ($n = 10$ independent biological replicates for all groups). *del-7(ok1187)* (light blue) contains a deletion for the intercellular region of DEL-7, and *del-7(gk688559)* (dark blue) contains an early stop codon. Note that *del-7* mutants display roaming behaviors, but that the reconstituted HySyn (dark and light red), as expected, is incapable of suppressing *del-7*, as *del-7* mutant animals are incapable of sensing food, incapable of activating NSM and therefore incapable of inducing the release of HyPep in the presence of food. This experiment demonstrates that the HySyn suppression of the *tph-1* mutants results from the activity-dependent release of HyPep from NSM upon encountering food. One-way ANOVA followed by a Tukey's multiple comparison post hoc test was used to compare the means of each group. * indicates $p < 0.05$, ns indicates that no statistically significant difference was observed. Error bars (black) represent the mean of each group and 95% confidence intervals. Schematics were made with BioRender[43]. Source data are provided as a Source Data file.

context-specific applications. In future studies, it will be useful to characterize HySyn dynamics in the context of tissue culture, slice preparation, and intact circuits using both mass spectrometry-based analyses of HyPep output and synthetic peptide-based analyses of HyCal activation dynamics.

The HySyn system is fully compatible with existing tools to optogenetically or chemogenetically control neural circuits but provides an innovative and complementary avenue to control the wiring of these circuits. As a volume-transmission neuromodulatory connection, the synthetic connections created by HySyn

could be applied in a wide range of neuroanatomical configurations, including in the absence of direct ultrastructural synaptic contact. Our engineered HySyn system, therefore, enables biasing or re-configuring neural circuits into discrete states for in vivo dissection of the role of neuromodulation to establish neural circuit logic and connectivity.

## Methods

**Molecular biology**. Optogenetic and calcium imaging plasmids were obtained from Addgene (ChRoME[22], 108902; Chrimson[22], 105447; GCaMP6f[24], 40755). All HySyn synaptic components were synthesized (Gene Universal, Newark DE USA) with flanking attB1/B2 sites for subsequent Invitrogen BP Gateway recombinational cloning into pDONR221 entry vector (Thermofisher, Waltham MA USA). To produce mammalian expression constructs, Gateway LR recombination was performed into the pEZY3 expression construct (Addgene 18672). For *C. elegans* expression, we used LR recombination subcloning with a Multisite Gateway system[7] to generate expression constructs. Core HySyn components are available from Addgene, and all plasmid information, including sequences, are listed in Supplementary Data 1. The primers used to genotype the mutant *C. elegans* strains are listed in Supplementary Data 3.

**Cell culture and transfection**. Neuro2a neuroblastoma cells (gift from Zhao-Wen Wang, UConn, Neuro-2a (ATCC CCL-131)) were cultured in Opti-MEM (ThermoFisher) supplemented with 5% fetal bovine serum (FBS, ThermoFisher) and penicillin/streptomycin. Cell transfection was performed with Lipofectamine 2000 (ThermoFisher) in Opti-MEM media according to the manufacturer's protocol. For electrophysiology experiments, ChRoME-mRuby2 fusion construct (Addgene 108902) was used to label cells co-transfected with HyPep and the optogenetic tool, whereas HyCal-expressing cells were labeled with pIRES2-GFP (ClonTech, PT3267-5). For calcium imaging experiments, a Chrimson-mRuby2 fusion construct (Addgene 105447) was used to label cells co-transfected with HyPep and the optogenetic tool, whereas HyCal-expressing cells were labeled with pCMV-GCaMP6f (Addgene 40755). Transfections were performed in separate dishes for the pre- and postsynaptic components. After ~24 h, transfected cells were dissociated with trypsin-EDTA (ThermoFisher) followed by mixing to co-culture the separately transfected populations on poly-L lysine-coated coverslips. Electrophysiology and calcium imaging were performed 24–48 h after co-culture. For electrophysiology experiments, Chrimson-mRuby fusion construct were used to label cells transfected with the optogenetic tool, whereas HyCal-expressing cells were labeled with pIRES2-GFP (ClonTech, PT3267-5).

**Electrophysiology and calcium imaging**. Coverslips with Neuro2a cells were mounted in the QE-1 chamber (Warner Instruments, Hamden CT USA) on an MT1000 stage (Sutter Instruments, Novato CA, USA) at room temperature under an Axio Vert.A1 microscope (Zeiss, Jena, Germany) equipped with filter set 63HE for mRuby and 38HE for GFP and GCaMP. Visualization and imaging of cells were performed with Obj. EC Plan-Neofluar ×5/0.16 M27 (420330-9901-000) or Obj. W Plan-Apochromat ×40/1.0 DIC M27 (421462-9900-000) on poly-L lysine-coated coverslips. Cells were bathed in an external solution containing (in mM): 140 NaCl, 1.3 KCl, 4 CsCl, 2 TEACl, 2 CaCl, 0.8 MgSO4, 5.5 D-glucose, 10 HEPES. Illumination for imaging and optogenetics was achieved using the Lambda-421 optical beam combiner (Sutter Instruments) using maximal output from the following LEDs: OBC-440, OBC-480, OBC-506, OBC-561, and OBC-590. Patch pipettes were pulled with a P1000 micropipette puller (Sutter Instruments) to 3-5 MΩ, then filled with an internal solution containing (in mM): 140 KCl, 1 MgCl₂, 5 K₄BAPTA, 3 CaCl₂, 25 HEPES. Electrophysiological data were acquired with the Double Integrated Patch Amplifier and Data Acquisition unit and SutterPatch software (Sutter Instruments). Cells were held at −70 mV for optogenetic experiments and all recordings were performed sequentially. Individual recordings of optogenetically evoked responses are shown in Supplementary Fig. 2. For calcium imaging, imaging and stimulation were achieved by alternate 500 ms exposures with the OBC-480 + 38HE and the OBC-590 + 63HE LED/filter combinations. In the "Light Off" condition (Fig. 2b, c, left panels), the OBC590 was turned off at the hardware switch. Solution transfer and washout experiments for calcium imaging were achieved manually with the aid of a micropipette in a total 1 ml bath solution. Calcium imaging, also on the Axio Vert.A1, was captured with an ORCA-Flash4.0 LT (Hamamatsu, Hamamatsu City Japan) controlled by μManager[9]. ROIs were manually generated around individual cells (and a reference background sample) in a maximum intensity projection using ImageJ (1.52)[42]. Mean ROI intensities were quantified for each frame and exported to MATLAB for image quantification and planned comparisons using the non-parametric Mann–Whitney–Wilcoxon test. Briefly, the background sample was first subtracted from each ROI, then a $\Delta F/F$ value was calculated for each frame, $F$, using the minimal fluorescence over the sample window as $F_0$ in the equation $(F - F_0)/F_0$. Because HySyn led to a gradual rise in GCaMP signal over time (Fig. 2c, f), we compared mean $\Delta F/F$ signals in the initial 2 min with those in the final 2 min of the recording interval to calculate a change in GCaMP signal ($\Delta F/F_{final} - \Delta F/F_{initial}$) in Fig. 2d, g.

**Worm transgenesis and behavior**. Transgenic lines were created by microinjection into the distal gonad syncytium as previously described[10] and selected based on the expression of co-injection markers, Punc-122::GFP or Punc-122::dsRed (Supplementary Data 1). Confocal images of transgene expression were acquired using Volocity (Perkin Elmer) on the UltraView VoX spinning disc confocal microscope with a NikonTi-E stand and a ×60 CFI Plan Apo VC, NA1.4, oil objective. Figures were prepared with FIJI[42], Adobe Illustrator (2020 24.3.0), and BioRender[43]. Animal migration on an agar pad was monitored for 30 min at 2 fps using a MightEx camera (BCE-B050-U). Trajectories were analyzed using an adaptation of the MagatAnalyzer software package as previously described[44]. Track analyses and planned comparisons using non-parametric Mann–Whitney–Wilcoxon test were implemented in MATLAB. A list of nematode strains and the corresponding genotypes used in this study can be found in Supplementary Data 2.

**Off-food exploration assays**. Synchronized young adult populations were washed in M9 buffer then transferred by pipette to the 20 °C behavioral test plate (22-cm × 22-cm agar plates). Worms were obliquely illuminated using an array of 624 nm LEDs and migration was monitored for 120 min at 2fps using a MightEx camera (BCE-B050-U). Animal speed was analyzed using LabView (2011, v11.0) and an adaptation of the MagatAnalyzer (v1.0) software package[44,45] and custom MATLAB scripts. Briefly, MagatAnalyzer uses a published and well-characterized machine vision approach to extract animal centroid position over time[7,44,45]. To estimate speed over the entire trajectory, as in Fig. 3I, we calculated the slope of a linear fit of the displacement for this centroid over time.

**On-food exploration assays**. Using methods adapted from prior work[30,32], 10 young adult animals (for each of the tested genotypes) were cultivated at 20 °C and picked to individual 60 mm NGM plates uniformly seeded with *E. coli* strain OP50. After 16 h, individual animals were removed from plates, and plates were superimposed on a 3.5 mm square grid. Using a dissecting scope, the number of squares containing a worm track (out of a maximum of 178) were counted. All genotypes were tested in parallel within the same 24 h period to account for any day-to-day variation in behaviors, and the scorer was blinded to the genotypes.

**Head thrashing assays**. L1-staged animals, with co-injection markers for both HyPep and HyCal, as well as appropriate tissue expression (pan-neuronal, body wall muscle, GABAergic neurons), were picked from an NGM plate uniformly seeded with *E. coli* strain OP50 and submerged into an NGM plate containing M9 buffer. Animals were allowed to swim for 30 s to remove any residual OP50 before being filmed for ~10 s using IC Capture Easy Image Acquisition software (Version 2.5.1525.3931, 64 bit) set to 30 fps. Swimming videos were filmed using a USB 3.0 Industrial Color Camera (Model# DFK 23UX236, The Imaging Source, LLC) attached to a Leica 165 at a 10.0 zoom. Animals were age-matched and scored at the L1 stage to eliminate possible effects from developmental arrest resulting from expression of the pan-neuronal reconstitution of HySyn (in rab-3::HySyn animals).

**Head thrashing analysis**. Worm behavior was tracked using DeepLabCut (version 2.2b8), a deep convolutional network that utilizes pretrained residual networks to robustly track animal behavior[28]. We labeled 10 videos using DeepLabCut's GUI toolbox with the following changes to default config file parameters: 8 evenly-spaced body parts from head to tail were labeled across 15 frames (set as "numframes2pick" value); label size was set to 2, alpha value to 0.7, p-cutoff to 0.9. Frames to be labeled were extracted using the *k*-means clustering function. Other default parameters for extracting frames were maintained. Following this step, the frames were labeled manually, a training set was created using the resnet_50 network and default augmentation method, and the network was trained with the default parameters outlined in prior work[29]. The resulting network was used to label all frames across each video and body part locations in each frame were saved for downstream analyses. To estimate thrashing, the head angle vector, described by the first two points along the body axis in the head and the neck, was quantified across all frames. After aligning to a common origin, the change in angle of this vector was calculated between successive frames captured at 30 fps. The median head angle change per second value was used to quantify individual animal thrashing. Illustration of angle quantification is available in Supplementary Movies 2 (full speed) and 3 (10× slower speed clip).

**Reporting summary**. Further information on research design is available in the Nature Research Reporting Summary linked to this article.

## Data availability

The source data generated in this study (plasmids and vectors) are available at Addgene (Deposit #78628). All relevant sequences are available in Supplementary Data 1. The data for the head thrashing assays (Supplementary Fig. 3a) is available on GitHub (https://doi.org/10.5281/zenodo.4782623). Source data are provided with this paper.

## Code availability

The custom code used for analysis in this study can be found at https://doi.org/10.5281/zenodo.4782623. Source data are provided with this paper.

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

## Acknowledgements

This work was initiated in the Grass Laboratory at the Marine Biological Laboratories (MBL) with funding through a Grass Fellowship awarded to J.D.H. Thanks to Richard Goodman (OHSU) for encouragement during the conceptualization of the fellowship application, and the 2019 Grass Fellows, Mel Coleman (Grass Director), and Christophe Dupré (Associate Director) for advice and support during the summer fellowship. We thank the MBL Division of Education and participants in the Vendor Equipment Loan Program. Special thanks to Sutter Instruments, who generously provided all electrophysiology equipment and substantial on-site assistance, and Zeiss, who provided on-site assistance at MBL. We thank Zhao-Wen Wang and Ping Liu (UConn) for guidance and training in patch-clamp electrophysiology, as well as providing Neuro2a cells. We thank Rob Steele (UCI) for supplying *Hydra*, as well as advice and inspiration on *Hydra* biology. We thank members of the Colón-Ramos lab and Hari Shroff (NIH) for thoughtful comments on the manuscript. We thank Michael Koelle and Andrew Olson (Yale University) for advice and reagents regarding serotonin rewiring experiments. We also thank Steve Flavell (MIT) for ideas and reagents regarding the experiments associated with *del-7*. We thank Life Science Editors for editing assistance. D.A.C.-R. is an MBL Fellow. Research in the D.A.C.-R. lab was supported by NIH R01NS076558, DP1NS111778, and by an HHMI Scholar Award.

## Author contributions

J.D.H was involved in the conceptualization of the HySyn tool, its design, and in the performance and analyses of all experiments. E.M.W. performed the HySyn reconstitution experiments in *C. elegans* in varying mutant backgrounds. T.S. performed the localization experiments in AIB. Z.D.K. performed the head thrashing analysis. All authors contributed to the preparation of the manuscript, with J.D.H. leading the writing efforts. D.A.C.-R. supervised the completion of the work and the manuscript.

## Competing interests

The authors declare no competing interests.
