## [Peer Review File · Nature Communications]

Reviewer #1 (Remarks to the Author):

The authors describe a molecular tool, which they call HySyn, consisting of a neuropeptide (which they call HyPep) and its receptor (which they call HyCal), a peptide ligand-gated calcium channel, both from Hydra, which had been previously characterized. The neuropeptide and receptor both seem to be orthogonal to most eukaryotic neuropeptides and receptors and thus can likely be used as molecular tools in model systems without unintentionally perturbing native physiology. Additionally, some engineering of the neuropeptide gene was done to allow for normal processing by pre-pro-convertase, etc.

They show in a neuron-like cell line, Neuro2A, in culture, by electrophysiology and calcium imaging, that stimulation of cells expressing HyPep leads to signals in cells expressing HyCal in the same culture. They also show that stimulation of the cells expressing HyPep followed by transfer of the media from that dish to a dish with cells expressing HyCal results in activation of the HyCal expressing cells in the second dish.

Finally, they demonstrate that HySyn works in *C. elegans*. Expression of HyPep in all neurons and HyCal in all muscles of the worm resulted in uncoordinated movement, reduced speed, and smaller movement tracks. Importantly, expression of HyPep in neurons alone (without HyCal), or expression of HyPep in intestines and HyCal in muscles did not change these movement parameters, and a mutant worm strain with disrupted neuropeptide processing also did not show the phenotype.

The concept of an exogenous, orthogonal released neuropeptide and corresponding receptor as an approach to modulate signaling in the nervous systems of model organisms is interesting. The data from HySyn in *C. elegans* are convincing that the HySyn transgenes are affecting physiology and behavior in vivo, likely via the proposed mechanism of HyPep release by neurons and binding at HyCal-expressing muscle cells resulting in disrupted muscle calcium signaling via opening of the HyCal channels.

However, there are numerous substantial issues with this manuscript.

General points:

HySyn is presented as a new molecular tool, but there is no discussion of the advantages and disadvantages of HySyn relative to other tools. A few examples that come to mind are Tango, Trans-TANGO, allostatin/allostatin receptor, chemogenetic receptors like DREADDs, optogenetic receptors. Discussion of how you might gate this system such that it is not always active, and the pros/cons of using calcium influx as a heterologous synapse would also be valuable. What effect would this system have on neurons that are spontaneously firing and have elevated calcium?

I honestly can't think of an example application where one would want to use HySyn relative to other existing tools with better specificity, better spatial or temporal resolution, or a more generalizable output like transcription. The authors mention using the neuronal HyPep/muscle HyCal worm movement phenotype to carry out a forward genetic screen to look for things involved in neuropeptide processing. This application made sense to me but is relatively narrow and I'm not sure if there are likely interesting things to be found (I don't know that field). Regardless, the authors don't actually carry out that screen. They similarly don't demonstrate any application that could not be done with previous tools, a standard bar for the introduction of an impactful new molecular tool.

Stylistically, throughout the manuscript, including the title and abstract, there is grandiose language about creating synthetic synapses and rewiring neural circuits. This is misleading with respect to what was actually done. Essentially, they perturbed all of the neuromuscular junctions in *C. Elegans* at the same time such that the worms couldn't move. In general, the manuscript contains numerous claims that are not strongly supported by data. The authors should stick to describing what was done and what was learned.

The experiments in cell culture are missing proper controls, have very low replicate numbers (apparently N=1 in some cases?), or both.

The methods are too succinct. It would be very difficult to reproduce these experiments in another laboratory.

Specific points:

-The manuscript states that the HySyn system is from the cnidarian Hydra, but several RFamides and receptors have been described from this organism. Please specify the sequence of the proposed RFamide after cleavage of the pre-pro-peptide and which Hydra Na⁺ Channel (HyNaC) receptor was used. Even better: additionally deposit the plasmids at Addgene along with the full sequences and reference that.

-“Expression of the HyPep synthetic pre-pro-peptide carrier in mammalian Neuro2A cells resulted in localization and transport of a fluorescent reporter to the expected intracellular compartments and vesicular release sites (Fig S1)”. Fig S1 shows an image of two cells. There is no co-localization of the labeled HyPep signal with existing markers of the cellular compartments they refer to or quantification of that localization.

-There is no data showing the expression or subcellular localization of the HyCal receptor in either cell culture or *C. elegans*. This makes it impossible to know which cells from Fig 2 express the HyCal receptor and would be expected to respond to the peptide.

-There is no direct demonstration that the proposed mature HyPep peptide is released by cells following stimulation. This could be done by mass spectrometry for example. Addition of synthetic HyPep peptide to HyCal-expressing cells and monitoring calcium activity would also help to show causality of mature neuropeptide release producing calcium signals. Functional perturbation of HyCal using known small molecule inhibitors to show that the response is specific (amiloride or diminazene).

-Fig 1d is labeled “untransfected control”, but several of the cells in the field of view from 1d are expressing the red fluorescent marker in panel 1b.

-Fig 1c-e This experiment appears to be N=1 as only one trace for each condition is shown and replicates are not described anywhere else. If true, that does not strongly support the conclusions drawn. If not true, please describe the sampling methods better.

-The experiments described in Fig 2 lack replicate numbers. Traces from individual cells are shown, but those could easily have been from one field of view of one replicate.

-“We observed that repeated optogenetic stimulation of the presynaptic HyPep-expressing

cells produced an integrating current in the postsynaptic HyCal receptor expressing cells (Fig 1h).” There is no Fig 1h.

-Fig 2d,g – the Y-axis is missing units

-The experiment described in Fig 2e-g (transferring media) is missing a negative control where media is collected from stimulated cells not expressing HyPep. Prior work has shown that neurons in culture can be robustly stimulated by only a media change.

-The plasmid used for GFP expression in the experiments shown in Fig 1 is not described.

-More details of the quantitation of GCaMP signal change in Fig 2d is necessary. The caption says (Final $\Delta F/F$ - Initial $\Delta F/F$), but the traces show that some cells are active in the middle of the timecourse but not at the beginning or end.

-Fig S1c is missing scale bars

-The promotor for driving intestinal expression of HyPep in *C. elegans* is not described.

-The error of the mean response from GCaMP6s can be plotted on the average traces shown in Fig 2.

-“We designed artificial neuropeptide spacers containing consensus cleavage sites (Fig 1b, red lines” Should read Fig S1b.

-“suppresses the neuromuscular HySyn paralysis phenotype (Supplementary Video S1, Fig 2c)” should read Fig 3c

- Scale bars and time stamps should be added to the supplemental video.

-Formatting of time: 4-min, 5min etc.

-Formatting of Neuro2A vs Neuro2a vs N2A cells.

-Formatting of neuromuscular vs. neuro-muscular.

Reviewer #2 (Remarks to the Author):

In their manuscript entitled “HySyn: A genetically-encoded synthetic synapse to rewire neural circuits in vivo”, J. D. Hawk and D. A. Colón-Ramos describe an approach to artificially establish a peptide-based communication between cells. This technique named “HySyn” uses a Hydra derived presynaptically neuropeptide and a matching postsynaptic cation channel, that is opened by the peptide. While triggering neuropeptide release could be achieved using channelrhodopsins, postsynaptic cation influx could be measured electrophysiologically and by calcium imaging. The present work introduces a potentially novel technique to control communication between cells and is a remarkable accomplishment arising from work of both authors being fellow at MBL Woods Hole.

However, while the novelty of this approach is claimed to be first of its kind, the authors commit the

negligent mistake to oversell their technique as rewiring and synapse forming, although they show that the communication established is volumetric. If by any means this manuscript should be recommended for publishing in Nature Communications, the revised version should unequivocally clarify that the approach neither leads to formation of new synapses nor can accomplish rewiring of neuronal networks. Hence, the authors should also rename their approach.

In addition the following issues should be solved:

Major issues:

1) As written above, no synapse formation is shown by HySyn. As the Authors demonstrate themselves, transmission between cells is volumetric and occurs by release of the neuropeptide (HyPep) in excited cells and binding of HyPep derived neuropeptides to the receiving receptor (HyCal) expressing cells. A synapse is always spatially specific. The authors should remove/replace all related text fragments that claim synapse formation or rewiring of cells/neurons aside from theoretical considerations.

2) While I fully understand that the authors did not include all the genetic / cloning information in their preprint variant of the MS, they definitely should have done by submitting to Nature Communications. It is neither clear how the constructs are composed completely nor where they are derived from, while source information is only partly available.

3) Figure 1 and related text: panel c,d,e are actually cut out from panel b and limited to consist of only one channel. This is misleading, e.g. in panel d are also cells that to express HyPep but only DIC is shown.

The optogenetically evoked photocurrents are comparably low in amplitude. What is the reason for this? It is not reported at which voltage cells have been clamped. In addition, there are no statistics given for any electrical response measured (only listed in the reporting file that it has been measured three times). Is there any statistics for responders/nonresponders as observed for the calcium imaging data?

Further, it is not clear whether currents have been measured sequentially or if the shown recordings are actually paired.

4) In figure 1, the HyCal receptor does not desensitize or inactivates, which is explained by the intrinsic properties of the used receptor. This is not useful for any application as reversibility would be required. Although the authors show that washing out the HyPep solution in the following experiments can get calcium response back to baseline, the authors should investigate how open HyCal can be deactivated / how unspecific proteases in neuronal tissue potentially terminate signaling with respect to further application

5) While the authors demonstrate that HyCal is receiving input from excited HyPep expressing cells by electrophysiology and calcium imaging, it is absolutely not clear whether HySyn can excite receiver cells e.g. if the technique can lead to “postsynaptic” action potential firing. Without showing the latter, the approach would only establish some communication between two neuronal populations that might influence or perturbate the receiving cells but not more than that.

6) For the *C. elegans* experiment, only continuous and non-specific signaling is demonstrated (muscle contraction or paralysis). Moreover, the worms appear to be smaller compared to WT or other (which is not quantified yet) indicating that continuous channel opening induces all kind of developmental issues (which would be the case for any constitutively open channel expressed in the

C. elegans muscle tissue).

7) It appears feasible that the presented HySyn approach could enable communication between two neuronal populations that are not naturally connected. However, this has not been demonstrated and would be the ultimate goal that should be achieved. Hence, such demonstration in slices or in vivo mouse experiment should be carried out.

Related to that, suitable constructs for neuronal expression/ virus preparation should be prepared. In line with this, the authors should include constructs that encode for channelrhodopsin fused or cleaved (P2A, IRES etc.) as for general application co-transfection or -transduction is cumbersome.

8) Figure 2 f,g: the authors excluded 25 of 34 total cells from the analysis as they were not responding. This is absolutely inappropriate.

9) At the end of the MS, the authors judge their approach based on the demands [1) to 5) described in the introduction of the MS]. However, this is again not done appropriate way, overselling and not objectively assessed. The authors should honestly discuss their achievements compared with the demands they requested themselves. There is a lot of space for improvements (partly discussed here).

10) The authors should validate that the neuropeptide released by HyPep expressing cells do not cause any activation of other receptors in mammals.

Minor issues:

1) Please provide line numbers during next round of revision.

2) A better visualization or labeling of the two different ChRs used within the figures would be beneficial (incl. blue light, orange light (Chrimson) etc.).

3) The experimental details are not fully given. The manuscript should at least contain enough details to enable repetition of the experiments by others. Aside from aforementioned missing details: e.g. illumination intensities, wavelengths, filters, integration times, temperature, calcium data correction (e.g. bleaching, normalization to expression levels?) etc. The authors should provide sufficient details.

4) Although the authors did not find a clear spatial pattern of activation with respect to the location of cells, calcium kinetics should be quantified in general.

5) Due to a broad readership, the authors should provide some introduction regarding neuronal communication of Hydra.

6) Figure numbering; The authors merged and renumbered figures and legends between their initial preprint and the submitted version of the MS. This is no pleasure to read. The supplemental figure is completely missing

Sincerely, Jonas Wietek

Reviewer #3 (Remarks to the Author):

This article presents a new method for engineering new chemical synapses in using genetically-encoded reagents. This involves the expression of a neuropeptide-gated cation channel from Hydra in the postsynaptic cell and its activating peptide ligand in the presynaptic cell. They demonstrate that this technique can work by expressing the components in cultured neuro2a cells and showing that optogenetic activation of the peptide expressing cell leads to currents or calcium transients in the channel-expressing cell. They also provide evidence that the technique can work in vivo by showing that *C. elegans* expressing the peptide in neurons and the channel in muscle show defective locomotion.

This is a really interesting idea and the data presented indicate that the technique has real potential. The data are a bit preliminary in places and in an ideal world there are a few things that could be done to more conclusively demonstrate that the method can really be used to engineer functional synapses in vivo. Obviously in these times it is not reasonable to expect much in the way of additional experiments though, so the comments below should be taken as suggestions of where data could be added if the authors have it or the text modified to highlight areas of uncertainty.

1. The authors conclude that their results "show the creation of a de novo synapse specifically through the reconstitution of the HySyn system". One question I had is, do they authors have evidence that the receptor is postsynaptically localized? Since the authors hypothesize that the peptide acts through volume transmission, in principle the receptor could be localized a considerable distance from the site of peptide release. This doesn't matter so much in the artificial culture system, but in vivo it seems significant where the receptor is localized. If the authors don't have data that bears on this, I think they should at least mention this as a possible issue for the use of the method in vivo.

2. A second question related to the use of this system to engineer synapses concerns the temporal dynamics. In culture, optogenetic activation of the channel leads to a current that does not inactivate. In vivo, such an engineered synapse would presumably stay active until the peptide is cleared. Do the authors have any idea what the timing of this would be? In principle they might test this by expressing the peptide in a neuron class that could be specifically activated (either by a sensory or optogenetic stimulus) and then observing the duration of the behavioral effects. If the authors have not done anything like this, they should at least discuss the issue as it is quite important for the use they envision for their technique.

3. Finally, a minor point: the authors state that the observed electrophysiology "is consistent with the known properties of both the receptor and neuromodulation by peptides". While the authors look at the effect of peptide released by cultured cells into the bath solution, I wonder if they have any data on the responses of receptor-expressing cells to purified peptide of known concentration? This would provide more precise information about the channel's expected properties in a heterologous system.

REVIEWER COMMENTS

Reviewer 1.

The authors describe a molecular tool, which they call HySyn, consisting of a neuropeptide (which they call HyPep) and its receptor (which they call HyCal), a peptide ligand-gated calcium channel, both from Hydra, which had been previously characterized. The neuropeptide and receptor both seem to be orthogonal to most eukaryotic neuropeptides and receptors and thus can likely be used as molecular tools in model systems without unintentionally perturbing native physiology. Additionally, some engineering of the neuropeptide gene was done to allow for normal processing by pre-pro-convertase, etc. They show in a neuron-like cell line, Neuro2A, in culture, by electrophysiology and calcium imaging, that stimulation of cells expressing HyPep leads to signals in cells expressing HyCal in the same culture. They also show that stimulation of the cells expressing HyPep followed by transfer of the media from that dish to a dish with cells expressing HyCal results in activation of the HyCal expressing cells in the second dish. Finally, they demonstrate that HySyn works in *C. elegans*. Expression of HyPep in all neurons and HyCal in all muscles of the worm resulted in uncoordinated movement, reduced speed, and smaller movement tracks. Importantly, expression of HyPep in neurons alone (without HyCal), or expression of HyPep in intestines and HyCal in muscles did not change these movement parameters, and a mutant worm strain with disrupted neuropeptide processing also did not show the phenotype. The concept of an exogenous, orthogonal released neuropeptide and corresponding receptor as an approach to modulate signaling in the nervous systems of model organisms is interesting. The data from HySyn in *C. elegans* are convincing that the HySyn transgenes are affecting physiology and behavior *in vivo*, likely via the proposed mechanism of HyPep release by neurons and binding at HyCal-expressing muscle cells resulting in disrupted muscle calcium signaling via opening of the HyCal channels. However, there are numerous substantial issues with this manuscript.

R1.1: HySyn is presented as a new molecular tool, but there is no discussion of the advantages and disadvantages of HySyn relative to other tools. A few examples that come to mind are Tango, Trans-TANGO, allostatin/allostatin receptor, chemogenetic receptors like DREADDs, optogenetic receptors. Discussion of how you might gate this system such that it is not always active, and the pros/cons of using calcium influx as a heterologous synapse would also be valuable. What effect would this system have on neurons that are spontaneously firing and have elevated calcium? I honestly can't think of an example application where one would want to use HySyn relative to other existing tools with better specificity, better spatial or temporal resolution, or a more generalizable output like transcription. The authors mention using the neuronal HyPep/muscle HyCal worm movement phenotype to carry out a forward genetic screen to look for things involved in neuropeptide processing. This application made sense to me but is relatively narrow and I'm not sure if there are likely interesting things to be found (I don't know that field). Regardless, the authors don't actually carry out that screen. They similarly don't demonstrate any application that could not be done with previous tools, a standard bar for the introduction of an impactful new molecular tool.

Thank you for highlighting the need for additional clarity in discussing and exemplifying how HySyn differs from prior methods, and its value and limitations. We addressed this concern in two ways:

- 1) We now better explain that HySyn is not intended to report synaptic connectivity or control whole-cell physiology (as most optogenetic or chemogenetic tools do). Instead, HySyn was designed as a tool to reconfigure connectivity through the creation of orthogonal peptidergic synapses that function through volumetric neurotransmission. We have edited the text accordingly, including the title, to communicate this more clearly. As suggested, we discuss the advantages of such a method, and its limitations, in the context of existing tools (lines 11-36; 75-77 and 292-306).

- 2) To better illustrate the utility of this tool, we add experimental data on the ability of HySyn to alter animal behavior (lines 224-274). Of note, we used HySyn to repair behavioral phenotypes in loss of function mutants by reconfiguring relationships between enteric sensory neurons and body wall muscles (new Fig 4). Briefly, serotonin acts as a neuromodulator to affect the locomotory state of animals. Serotonin is released from an enteric sensory cell which senses the presence of ingested food (bacteria) and secretes serotonin. Normally, in the presence of food, serotonin is secreted and animals change their locomotory strategy from “roaming” to “dwelling”. In mutants of the serotonergic biosynthesis pathway, animals remain in a state of roaming even in the presence of food. We tested whether targeted expression of HySyn to reconnect the enteric sensory neuron NSM to muscles could suppress this phenotype in serotonin mutants. Indeed, our use of HySyn to reconstitute a modulatory synapse artificially links the food-sensing neuron with muscles, recovering a “dwelling-like” phenotype when the HySyn-expressing animals are in the presence of food.

We then used *C. elegans* genetics to probe the specificity of the observed effect. For example, *del-7* mutants, which lack an NSM neuron-specific channel that senses the presence of food, also show abnormal roaming, as this mutation leaves the NSM neuron incapable of being activated on food, and releasing serotonin. Unlike the serotonin biosynthesis mutants, HySyn reconstitution in *del-7* mutants is incapable of suppressing the *del-7* phenotype, as we predicted, due to the fact that NSM is inactive and incapable of releasing the HyPep in the presence of food. As we now explain in the text, these types of experiments, which use orthogonal synthetic synapses to recover neuromodulatory signaling, are not possible with other existing tools in neuroscience.

Lastly, we include additional data in which we target HyPep and HyCal components to varying tissues, reconstituting sensory to NMJ neuromodulatory connections and characterizing the effects of these configurations in animal locomotion (new Fig S3a,b, new Supplementary Videos 2 and 3, discussed in lines 190-210).

R1.2. Stylistically, throughout the manuscript, including the title and abstract, there is grandiose language about creating synthetic synapses and rewiring neural circuits. This is misleading with respect to what was actually done. Essentially, they perturbed all of the neuromuscular junctions in *C. elegans* at the same time such that the worms couldn't move. In general, the manuscript contains numerous claims that are not strongly supported by data. The authors should stick to describing what was done and what was learned.

We have edited the manuscript, including the title and the abstract, to better communicate that the HySyn tool reconstitutes “synthetic modulatory neurotransmission” (instead of using the term “synthetic synapse”, as we did in the original manuscript). We also edited the text to communicate that when we discuss “synapses”, we refer to a peptidergic modulatory synapse (lines 44-47). We added text to explain that we purposefully designed a tool that alters neuromodulation as a way of extending the utility of HySyn (to modulate neuronal relationships that are not necessarily next to each other), and as a way of exploring neuromodulation over the endogenous circuit signaling pathways (lines 37-54; 75-77; 275-306).

R1.3. The experiments in cell culture are missing proper controls, have very low replicate numbers (apparently N=1 in some cases?), or both.

We now added additional traces and quantification for replicates of electrophysiological experiments in new Supplementary Fig S2.

R1.4. The methods are too succinct. It would be very difficult to reproduce these experiments in another laboratory.

We have now extended the methods, and added plasmid information and sequences in Supplementary Table 1. We have made all key plasmids available in Addgene.

R1.5. The manuscript states that the HySyn system is from the cnidarian *Hydra*, but several RFamides and receptors have been described from this organism. Please specify the sequence of the proposed RFamide after cleavage of the pre-pro-peptide and which *Hydra* Na⁺ Channel (HyNaC) receptor was used. Even better: additionally deposit the plasmids at Addgene along with the full sequences and reference that.

We edited the text to clarify that HySyn is derived from the RFamide-related peptide HyRFamide I/II, and the cognate receptor HyNAC 2/7/9 (lines 60-62). The constructs for the HySyn components have now been submitted to Addgene. A new Supplemental Table was created (Supplementary Table 1) to show all plasmids, with sequence information used in this study.

R1.6. “Expression of the HyPep synthetic pre-pro-peptide carrier in mammalian Neuro2A cells resulted in localization and transport of a fluorescent reporter to the expected intracellular compartments and vesicular release sites (Fig S1)”. Fig S1 shows an image of two cells. There is no co-localization of the labeled HyPep signal with existing markers of the cellular compartments they refer to or quantification of that localization. There is no data showing the expression or subcellular localization of the HyCal receptor in either cell culture or *C. elegans*. This makes it impossible to know which cells from Fig 2 express the HyCal receptor and would be expected to respond to the peptide.

We addressed the concern in two ways:

1) We have performed co-localization experiments in a polarized neuron of *C. elegans* called AIB^{1, 2, 31-3} (new Fig 3a-i). These experiments demonstrated that HyPep colocalizes with the dense core vesicle marker IDA-1/Phogrin in the presynaptic region of the neurite, whereas HyCal is enriched near the glutamate receptor GLR-1 in the postsynaptic region of the neurite. The new data are discussed in lines 175-189.

2) We edited the text for clarity, and it now states “This observation is consistent with our hypothesis that the engineered HyPep carrier harnesses the universality of the neuropeptide processing pathway to target the *Hydra* neuropeptide processing, transport and release” (lines 97-100). We also better explain in the text that in the cell culture experiments shown in Fig 2, co-transfection was performed to allow identification of cells transfected with the receptor.

R1.7. There is no direct demonstration that the proposed mature HyPep peptide is released by cells following stimulation. This could be done by mass spectrometry for example. Addition of synthetic HyPep peptide to HyCal-expressing cells and monitoring calcium activity would also help to show causality of mature neuropeptide release producing calcium signals. Functional perturbation of HyCal using known small molecule inhibitors to show that the response is specific (amiloride or diminazene).

Thank you for this comment. Due to the pandemic and in the context of the facilities available for research to our lab during this time, we could not perform the indicated experiments as suggested. To address this concern, however, we added a reference to prior work showing both small-molecule perturbation and peptide activation of the HyCal receptor configuration used in this study (in lines 279-282) and acknowledge the importance of these future experiments in the discussion (in lines 282-291).

R1.8. Fig 1d is labeled “untransfected control”, but several of the cells in the field of view from 1d are expressing the red fluorescent marker in panel 1b.

We edited the text and figure for clarity. The panel was intended to highlight the distinct cell types that are examined within a single experiment, where ‘untransfected control’ cells are those that have not been

transfected with either HySyn component (confirmed by the lack of the cotransfection marker). We have now edited the Fig 1 panel for clarity, and explain this point in the Methods and Figure legends.

R1.9. Fig 1c-e This experiment appears to be N=1 as only one trace for each condition is shown and replicates are not described anywhere else. If true, that does not strongly support the conclusions drawn. If not true, please describe the sampling methods better.

Initially we showed a representative sample for each condition. We have now added an additional figure, new Fig S2, where we provide traces and quantification for replicates of electrophysiological experiments.

R1.10. The experiments described in Fig 2 lack replicate numbers. Traces from individual cells are shown, but those could easily have been from one field of view of one replicate.

Replicates for the electrophysiology experiments corresponding to Fig 1 are now shown in new Fig S2. In Fig 2, GCaMP signal was monitored from individual cells, as shown in the diagram, and each individual cell is considered a replicate in this context. We now better explain this in the Methods (lines 373-391).

R1.11. “We observed that repeated optogenetic stimulation of the presynaptic HyPep-expressing cells produced an integrating current in the postsynaptic HyCal receptor expressing cells (Fig 1h).” There is no Fig 1h.

We have corrected this and carefully edited figure correspondence in this version of the manuscript.

R1.12. Fig 2d,g – the Y-axis is missing units

We have updated Fig 2 to include units in the Y-axis.

R1.13. The experiment described in Fig 2e-g (transferring media) is missing a negative control where media is collected from stimulated cells not expressing HyPep. Prior work has shown that neurons in culture can be robustly stimulated by only a media change.

We thank you for this comment. We did not carry out this experiment, due to the pandemic and current capabilities of our lab. Our experiments in Fig 2b illustrate that light stimulation in the absence of HyCal receptor does not evoke calcium transients and the washout experiments in Fig 2f illustrate that bath solution transfer alone was not sufficient to evoke calcium rises. While this requested experiment is a reasonable extension of the already performed controls, we have since updated the manuscript with several additional *in vivo* demonstrations that strongly suggest that without the presence of HyPep, the HyCal receptor is not activated by endogenous signals (new Fig 4 and new S3), and edited the text to discuss this appropriately.

R1.14. The plasmid used for GFP expression in the experiments shown in Fig 1 is not described.

We have updated the manuscript with an additional table, Supplementary Table S1, where we provide descriptions of all plasmids and their respective sequences, as well as their repository numbers in Addgene. Specifically, the plasmid used for HyCal-GFP expression in Supplementary Table S1 was pCMV, noted in Supplementary Table 1 (Addgene #'s 160418, 160419, 160420).

R1.15. More details of the quantitation of GCaMP signal change in Fig 2d is necessary. The caption says (Final $\Delta F/F$ - Initial $\Delta F/F$), but the traces show that some cells are active in the middle of the timecourse but not at the beginning or end.

We have now added the following text to the methods section to describe our GCaMP quantification: “Briefly, the background sample was first subtracted from each ROI, then a $\Delta F/F$ value was calculated for each frame, F , using the minimal fluorescence over the sample window as F_0 in the equation $(F-F_0)/F_0$. Because HySyn led to a gradual rise in GCaMP signal over time (Fig 2c & 2f), we compared mean $\Delta F/F$

signals in the initial 2 minutes with those in the final 2 minutes of the recording interval to calculate a change in GCaMP signal ($\Delta F/F_{\text{final}} - \Delta F/F_{\text{initial}}$) in Fig 2d & 2g.” in lines 395-400.

R1.16. Fig S1c is missing scale bars.

Thanks, we have added a scale bar to the Supplemental Figure S1c.

R1.17. The promoter for driving intestinal expression of HyPep in *C. elegans* is not described.

We have now added detailed description of plasmids in a new supplementary table (Supplementary Table 1) (as mentioned in the Methods section (lines 350-351), which reads “Core HySyn components are available from Addgene, and all plasmid information, including sequences, are listed in Supplementary Table 1.”)

R1.18. The error of the mean response from GCaMP6s can be plotted on the average traces shown in Fig 2.

We have since updated Fig 2 to include error of the mean response from GCaMP6f in panels b, c and f.

R1.19. “We designed artificial neuropeptide spacers containing consensus cleavage sites (Fig 1b, red lines)” Should read Fig S1b.

We have corrected this within the manuscript text and carefully edited figure correspondence in this version of the manuscript.

R1.20. “suppresses the neuromuscular HySyn paralysis phenotype (Supplementary Video S1, Fig 2c)” should read Fig 3c.

We have corrected this within the text and updated it to reflect the current location of this data, which now appears in Fig 3l.

R1.21. Scale bars and time stamps should be added to the supplemental video.

We have now added time stamps and scale bars to the original supplementary video, and to the new videos generated for this revision.

R1.22. Formatting of time: 4-min, 5min etc. Formatting of Neuro2A vs Neuro2a vs N2A cells. Formatting of neuromuscular vs. neuro-muscular.

We have now made the formatting suggestions throughout the manuscript, thank you.

Reviewer 2:

In their manuscript entitled “HySyn: A genetically-encoded synthetic synapse to rewire neural circuits in vivo”, J. D. Hawk and D. A. Colón-Ramos describe an approach to artificially establish a peptide-based communication between cells. This technique named “HySyn” uses a Hydra derived presynaptically neuropeptide and a matching postsynaptic cation channel, that is opened by the peptide. While triggering neuropeptide release could be achieved using channelrhodopsins, postsynaptic cation influx could be measured electrophysiologically and by calcium imaging. The present work introduces a potentially novel technique to control communication between cells and is a remarkable accomplishment arising from work of both authors being fellow at MBL Woods Hole. However, while the novelty of this approach is claimed to be first of its kind, the authors commit the negligent mistake to oversell their technique as rewiring and synapse forming, although they show that the communication established is volumetric. If by any means this

manuscript should be recommended for publishing in Nature Communications, the revised version should unequivocally clarify that the approach neither leads to formation of new synapses nor can accomplish rewiring of neuronal networks. Hence, the authors should also rename their approach. In addition the following issues should be solved:

R2.1. As written above, no synapse formation is shown by HySyn. As the Authors demonstrate themselves, transmission between cells is volumetric and occurs by release of the neuropeptide (HyPep) in excited cells and binding of HyPep derived neuropeptides to the receiving receptor (HyCal) expressing cells. A synapse is always spatially specific. The authors should remove/replace all related text fragments that claim synapse formation or rewiring of cells/neurons aside from theoretical considerations.

Thank you for this comment. We have updated the text, including the title and the abstract, to better reflect our intended use of the word 'synapse' to refer to a peptidergic modulatory synapse (explained in lines 44-47). We added a paragraph to explain that this configuration was favored as a way of extending the utility of the tool to modulate circuits that can be at a distance, and as a way of exploring neuromodulation over the endogenous signaling pathways (lines 37-54). Additionally, as described for R1.1, we have added data demonstrating the value of HySyn to reconstitute neuromodulatory relationships between cells *in vivo* (new Fig 4 and corresponding discussion in the manuscript).

R2.2. While I fully understand that the authors did not include all the genetic / cloning information in their preprint variant of the MS, they definitely should have done by submitting to Nature Communications. It is neither clear how the constructs are composed completely nor where they are derived from, while source information is only partly available.

Thank you for highlighting this unintentional omission. We have now added detailed description of plasmids in a new supplementary table (Supplementary Table 1) and made the key reagents publicly available via Addgene.

R2.3a. Figure 1 and related text: panel c,d,e are actually cut out from panel b and limited to consist of only one channel. This is misleading, e.g. in panel d are also cells that to express HyPep but only DIC is shown.

Thank you for highlighting the need for additional clarification. The panel was intended to highlight the distinct cell types that were examined within a single experiment, where 'untransfected control' cells are those that have not been effectively transfected with either HySyn component. Of note in this experiment, cells which were transfected with HyPep were co-cultured with cells transfected with HyCal to examine their interaction via HySyn reconstitution. We now better explain this in the text, legends and Methods (lines 356-368). We also added clarification of the relationship between Fig 1 (panel b) and cells in panels c-e, as well as arrows marking the specific cell in both images, as to orient the readers.

R2.3b The optogenetically evoked photocurrents are comparably low in amplitude. What is the reason for this? It is not reported at which voltage cells have been clamped. In addition, there are no statistics given for any electrical response measured (only listed in the reporting file that it has been measured three times). Is there any statistics for responders/nonresponders as observed for the calcium imaging data? Further, it is not clear whether currents have been measured sequentially or if the shown recordings are actually paired.

We have now updated the methods to indicate that cells were held at -70mV for optogenetic experiments and all recordings were sequential (lines 373-391). We have also now quantified all currents (new Fig S2) with optogenetically evoked currents ranging from 31pA to 111pA. These results are lower than those originally reported for this construct, ChRoME (530±/50pA)⁴. This difference is not surprising given the distinct construct delivery method (lipofection vs viral transduction) and cell type (Neuro2A vs pyramidal neuron) used in our study. We now show individual recordings (Fig S2c-e) to allow contrasting between the uniform observation of optogenetic responses in ChRoME-expressing cells (Fig S2c) compared to the

~50% electrophysiological response rate in the HySyn coupled cells (Fig S2e), consistent with our results with calcium imaging.

R2.4. In figure 1, the HyCal receptor does not desensitize or inactivates, which is explained by the intrinsic properties of the used receptor. This is not useful for any application as reversibility would be required. Although the authors show that washing out the HyPep solution in the following experiments can get calcium response back to baseline, the authors should investigate how open HyCal can be deactivated / how unspecific proteases in neuronal tissue potentially terminate signaling with respect to further application

As the reviewer indicates, our non-inactivating currents (Fig 1) and rapid washout (Fig 2) point to peptide release and stability as the key factors for determining the duration of neuromodulation by HySyn. We have now addressed the *in vivo* kinetics of HySyn behavioral effects (new Fig S3d,e). Briefly, we genetically targeted the HyPep to a food-inducible enteric neuron, NSM, and targeted the HyCal receptor to the body wall muscles (pNSM::HySyn). These animals display a state of decreased locomotion (dwelling) when they are reared and maintained within the presence of food (new Fig 4 and new S3c). To examine the inactivation kinetics of HySyn in this context, and to address the comment above, we conducted an experiment where we measured the locomotory speed of pNSM::HySyn animals immediately after they are removed from food (which result in inactivation of the food sensing neuron in which HyPep is expressed, and presumably lack of transmission of HyPep; Fig S3d,e). The results showed a restoration of locomotion speed to wild-type levels in 70 min, which strongly suggests a desensitization of muscle-HyCal within 70 minutes of HyPep not being released by the food-sensing neuron. These *in vivo* kinetics are similar to those seen for other neuromodulators, such as vasopressin⁵.

Lastly, we have included text discussing the likely regulation of HyPep by extracellular proteases, a common source for limiting neuromodulatory activity (lines 271-274) and discussions of pharmacological inactivation of the HyCal receptor for tighter temporal control (lines 286-291).

R2.5. While the authors demonstrate that HyCal is receiving input from excited HyPep expressing cells by electrophysiology and calcium imaging, it is absolutely not clear whether HySyn can excite receiver cells e.g. if the technique can lead to “postsynaptic” action potential firing. Without showing the latter, the approach would only establish some communication between two neuronal populations that might influence or perturbate the receiving cells but not more than that.

We did not examine action potential firing. To acknowledge this, we discuss it as a limitation of the study (in lines 301-306). However, we now show *in vivo* evidence on how HySyn can be used to rewire neuronal relationships through peptidergic neurotransmission, and examine these relationships in the context of varying mutant backgrounds that affect behavior, as explained in R2.4 and R1.1 and shown in new Fig 4.

R2.6. For the C. elegans experiment, only continuous and non-specific signaling is demonstrated (muscle contraction or paralysis). Moreover, the worms appear to be smaller compared to WT or other (which is not quantified yet) indicating that continuous channel opening induces all kind of developmental issues (which would be the case for any constitutively open channel expressed in the C. elegans muscle tissue).

To further demonstrate HySyn's ability to alter animal behavior via reconfiguration of neural circuits, we have conducted additional experiments which now appear in new Fig 4 and new Fig S3 (and which we explain above in the response to Reviewer 1, Comment 1).

We also provide *in vivo* evidence of the extinction of the HySyn effect (Fig S3d,e; explained above in response to Reviewer 2 Comment 4).

R2.7. It appears feasible that the presented HySyn approach could enable communication between two neuronal populations that are not naturally connected. However, this has not been demonstrated and would be the ultimate goal that should be achieved. Hence, such demonstration in slices or in vivo mouse experiment should be carried out. Related to that, suitable constructs for neuronal expression/ virus preparation should be prepared. In line with this, the authors should

include constructs that encode for channelrhodopsin fused or cleaved (P2A, IRES etc.) as for general application co-transfection or -transduction is cumbersome.

Thank you for highlighting the need to further demonstrate HySyn's ability to alter the communication between distinct populations of neurons. We have since updated the manuscript with several different instances of this, and provide further description of the behavioral modifications we achieved in the New fig 4, as discussed for Reviewer 1, Comment 1.

Regarding the specific suggested experiments of slices and *in vivo* mouse studies, we were unable to perform them in the context of the pandemic, but now acknowledge their value in the discussion of the manuscript (lines 303-306).

R2.8. Figure 2 f,g: the authors excluded 25 of 34 total cells from the analysis as they were not responding. This is absolutely inappropriate.

I think the reviewer is referring to Fig 2c and d, but for both Fig 2f and Fig 2b-c, all analyzed cells are shown. In the text and in the figure legends we indicate the number of responding cells for the analyses. In Fig 2g, we further analyze how the responding cells change based on the addition of HyPep, or washout, and in that context, select the 9 responding cells to examine the kinetics of the cellular responses. To further clarify this, we edited the figure legend (lines 504-506).

R2.9. At the end of the MS, the authors judge their approach based on the demands [1) to 5) described in the introduction of the MS]. However, this is again not done appropriate way, overselling and not objectively assessed. The authors should honestly discuss their achievements compared with the demands they requested themselves. There is a lot of space for improvements (partly discussed here).

We have added additional text to better compare and contrast the value of HySyn, including its limitations, at lines 292-306.

R2.10. The authors should validate that the neuropeptide released by HyPep expressing cells do not cause any activation of other receptors in mammals.

We were unable, in the context of the pandemic, to perform experiments in mammals. However, to better examine this point in *C. elegans*, we now include new data demonstrating that expression of just HyPep pan-neuronally (new Fig 3j, and new Supplementary Video 2) does not have any effect on development, behavior, animal locomotion (new Fig 3i, green) or head thrashing (new Fig S3a, green), which strongly suggests that HyPep does not cause activation of other endogenous receptors in *C. elegans*. We also now add additional discussion of these data and our previous data demonstrating that mouse neuroblastoma cells lacking the HyCal receptor fail to be activated by HyPep (Fig 2b). Discussed in lines 134; 196-199.

R2.11. Please provide line numbers during next round of revision.

We have now added line numbers and page numbers to the text.

R2.12. A better visualization or labeling of the two different ChRs used within the figures would be beneficial (incl. blue light, orange light (Chrimson) etc.).

Thank you for this recommendation. We updated the schematics in Fig 1 and S2 to represent the wavelength of light used for these optogenetic experiments.

R2.13. The experimental details are not fully given. The manuscript should at least contain enough details to enable repetition of the experiments by others. Aside from aforementioned missing details: e.g. illumination intensities, wavelengths, filters, integration times, temperature, calcium data correction (e.g. bleaching, normalization to expression levels?) etc. The authors should provide sufficient details.

We have edited the methods section for clarity and the requested specifics.

R2.14. Although the authors did not find a clear spatial pattern of activation with respect to the location of cells, calcium kinetics should be quantified in general.

Thank you for this comment. We did not quantify the calcium kinetics of this particular experiment, but now acknowledge this limitation and clarify that the expected absence of a clear spatial activation pattern was consistent with our electrophysiology experiments, and that this is consistent with the known properties of neuromodulators (in lines 142-145).

R2.15. Due to a broad readership, the authors should provide some introduction regarding neuronal communication of *Hydra*.

Thank you for this comment. We have now added text in which we describe briefly introduce neuronal communication of *Hydra* (in lines 55-56).

R2.16. Figure numbering; The authors merged and renumbered figures and legends between their initial preprint and the submitted version of the MS. This is no pleasure to read. The supplemental figure is completely missing

Thank you for bringing this to our attention, we have corrected it.

Reviewer #3

This article presents a new method for engineering new chemical synapses in using genetically-encoded reagents. This involves the expression of a neuropeptide-gated cation channel from Hydra in the postsynaptic cell and its activating peptide ligand in the presynaptic cell. They demonstrate that this technique can work by expressing the components in cultured neuro2a cells and showing that optogenetic activation of the peptide expressing cell leads to currents or calcium transients in the channel-expressing cell. They also provide evidence that the technique can work in vivo by showing that *C. elegans* expressing the peptide in neurons and the channel in muscle show defective locomotion. This is a really interesting idea and the data presented indicate that the technique has real potential. The data are a bit preliminary in places and in an ideal world there are a few things that could be done to more conclusively demonstrate that the method can really be used to engineer functional synapses in vivo. Obviously in these times it is not reasonable to expect much in the way of additional experiments though, so the comments below should be taken as suggestions of where data could be added if the authors have it or the text modified to highlight areas of uncertainty.

R3.1. The authors conclude that their results "show the creation of a *de novo* synapse specifically through the reconstitution of the HySyn system". One question I had is, do the authors have evidence that the receptor is postsynaptically localized? Since the authors hypothesize that the peptide acts through volume transmission, in principle the receptor could be localized a considerable distance from the site of peptide release. This doesn't matter so much in the artificial culture system, but *in vivo* it seems significant where the receptor is localized. If the authors don't have data that bears on this, I think they should at least mention this as a possible issue for the use of the method *in vivo*.

To address this concern, we include new *in vivo* data in *C. elegans* in which we carefully examined the subcellular localization of the HySyn components in single, polarized neurons that enable us to characterize the pre- or postsynaptic localization of the HyPep and HyCal. We observe that HySyn is enriched in presynaptic regions of neurons, where it co-localizes with dense-core vesicle marker IDA-1::mCherry, while the HyCal receptor is enriched in postsynaptic regions of the neurite, where it co-localizes with the postsynaptic receptor, GLR-3¹⁻³. We include this new data in Fig 3 and discuss it in the text (lines 175-189).

R3.2. A second question related to the use of this system to engineer synapses concerns the temporal dynamics. In culture, optogenetic activation of the channel leads to a current that does not inactivate. In vivo, such an engineered synapse would presumably stay active until the peptide is cleared. Do the authors have any idea what the timing of this would be? In principle they might test this by expressing the peptide in a neuron class that could be specifically activated (either by a sensory or optogenetic stimulus) and then observing the duration of the behavioral effects. If the authors have not done anything like this, they should at least discuss the issue as it is quite important for the use they envision for their technique.

Thank you for this comment, which prompted us to carry out additional experiments as suggested. Specifically, upon demonstrating that we could suppress the behaviors associated with loss of serotonin by expressing HySyn to “connect” a serotonergic food-sensing neuron and muscles, we reasoned, because the food-sensing neuron (called NSM) was selectively active in the presence of food, we could examine the extinguishing effects of HySyn in animals in which HyPep release was reduced upon removal from food (and NSM inactivation). We observed that the effect of HySyn (decreased movement, or “dwelling”), in pNSM::HySyn animals was slowly extinguished in a period of 70 minutes off food, consistent with its role as a neuromodulator, and consistent with the inactivation of the receptor when HyPep is not continually present. The dynamics we observed of behavior extinction *in vivo* are consistent to those observed for other neuropeptides, such as vasopressin⁵, now also discussed in the text (lines 269-271).

R3.3. Finally, a minor point: the authors state that the observed electrophysiology "is consistent with the known properties of both the receptor and neuromodulation by peptides". While the authors look at the effect of peptide released by cultured cells into the bath solution, I wonder if they have any data on the responses of receptor-expressing cells to purified peptide of known concentration? This would provide more precise information about the channel's expected properties in a heterologous system.

Thank you for this comment. While we were unable to perform this specific experiment in the context of the pandemic, we edited the text to acknowledge its importance: We now cite that the HyCal receptor has been expressed heterologously in *Xenopus* oocytes, where direct peptide inactivation leads to dose-dependent, non-inactivating currents, acknowledge we did not perform these experiments in this manuscript and highlight the value of these pharmacological characterizations in future experiments (lines 279-291).

References

1. White JG, Southgate E, Thomson JN, Brenner S. The structure of the nervous system of the nematode *Caenorhabditis elegans*. *Philos Trans R Soc Lond B Biol Sci* **314**, 1-340 (1986).
2. Sengupta T, *et al.* A neurite-zippering mechanism, mediated by layer-specific expression of IgCAMs, regulates synaptic laminar specificity in the *C. elegans* nerve ring neuropil. *BioRxiv*, (2020).
3. Moyle MW, *et al.* Structural and developmental principles of neuropil assembly in *C. elegans*. *BioRxiv*, (2020).
4. Mardinly AR, *et al.* Precise multimodal optical control of neural ensemble activity. *Nat Neurosci* **21**, 881-893 (2018).
5. Mens WBJ, Witter A, Van Wimersma Greidanus TB. Penetration of neurohypophyseal hormones from plasma into cerebrospinal fluid (CSF): Half-times of disappearance of these neuropeptides from CSF. *Brain Research* **262**, 143-149 (1983).

Reviewer #1 (Remarks to the Author):

In general, the revised manuscript is much improved. We had the following (minor) comments after reading the revised draft:

It would be beneficial to the reader for the figures to start with an introductory cartoon of the conceptual idea, rather than experiment number 1 being panel 1 in figure 1.

What is the tph worm mutant? I didn't find a description.

Figure Labeling in FigS3e and Fig S3d are swapped. In the figure e is the time trace and d is the 30 min binning. In the figure text d is the time trace and e is the 30 min binning.

The names in the supp video 2 do not match the figure panel S3a (they are the same thing, but it makes it unclear for somebody not in the field : Pan neuronal vs rab-3 and muscle vs. myo-cal) - nor are the different names explained in the figure legend.

Reviewer #2 (Remarks to the Author):

The authors greatly improved the quality of the manuscript by considering the suggestions and concerns given by all reviewers and provide additional data to support their claims and statements.

Although not all requests for additional experiments could not be fulfilled, the presented data is sufficient to publish the manuscript, especially with regard to the difficulties caused by the pandemic.

However, there is still one major point that has to be addressed. Although the authors tried to improve their use of language for description of the signal transmission between cells using their HySyn system, there are many cases where the reader is still kind of misled. As the authors correctly claim they established a peptidergic volume transmission that is able to induce neuromodulation between different cells. In that context using the term “rewire/rewiring” is already problematic. The same accounts for using the term “synapse” or “peptidergic synapse” in this context. A synapse is always defined spatially. Although peptidergic synapses exist, the authors do establish de novo peptidergic transmission and neuromodulation but no formation of any de novo synapses. The authors should thoroughly revise the use of the terms within their manuscript

The use of the terms presynaptic and postsynaptic for Neuro2a cells in culture should be done consequently using quotation marks as already done at some parts of the text (this also includes the figure legend). Alternatively, cells could be categorized as “sender” and “receiver” depending on which construct of the HySyn system they are expressing.

Line 110: I suggest to not use the term neuron for Neuro2a cells.

Fig.1 b,c,f: It is recommended to use perceptually uniform colormaps instead of “rainbow” colormaps, especially to make the heatmaps barrier free to visually impaired readers. Useful colormaps can be found here for instance: <https://hauselin.github.io/colorpalettejs/>

Line 272: It's "Jena" not "Jenna".

Sincerely, Jonas Wietek

Reviewer #3 (Remarks to the Author):

I think the authors have done a good job responding to reviewer comments. I particularly think it is noteworthy that they have provided a lot of additional experimental data which would not have been easy during the pandemic. I am in favor of publication.

APRIL 2021 REVIEWER COMMENTS HAWK ET AL

Reviewer #1:

In general, the revised manuscript is much improved. We had the following (minor) comments after reading the revised draft:

R1.1: “It would be beneficial to the reader for the figures to start with an introductory cartoon of the conceptual idea, rather than experiment number 1 being panel 1 in figure 1.”

We updated Figure 1, panel ‘a’ (left) with a cartoon illustrating the conceptual idea of HySyn, as suggested, and noted this in the text (line 54) and figure caption (lines 461-463).

R1.2: “What is the tph worm mutant? I didn't find a description.”

We have modified the text (lines 231-234) to introduce the ‘tph-1 mutant’ as an allele of tryptophan hydroxylase which affects serotonin biosynthesis.

R1.3: “Figure Labeling in FigS3e and Fig S3d are swapped. In the figure e is the time trace and d is the 30 min binning. In the figure text d is the time trace and e is the 30 min binning.”

Thank you for bringing this to our attention. We have fixed this error.

R1.4: “The names in the supp video 2 do not match the figure panel S3a (they are the same thing, but it makes it unclear for somebody not in the field : Pan neuronal vs rab-3 and muscle vs. myocal) - nor are the different names explained in the figure legend.”

To address this comment, we have done two things:

- 1) We have updated the title slide in Supplementary Video 2 with the names of the tissues in which HySyn components are being expressed.
- 2) We have updated the figure caption for Supplementary Video 2 (lines 675-680) such that the names of the tissues, along with the promoters used to drive expression in those tissues, are listed side-by-side.

Reviewer #2:

The authors greatly improved the quality of the manuscript by considering the suggestions and concerns given by all reviewers and provide additional data to support their claims and statements.

Although not all requests for additional experiments could not be fulfilled, the presented data is sufficient to publish the manuscript, especially with regard to the difficulties caused by the pandemic.

R2.1: “However, there is still one major point that has to be addressed. Although the authors tried to improve their use of language for description of the signal transmission between cells using their HySyn system, there are many cases where the reader is still kind of misled. As the authors correctly claim they established a peptidergic volume transmission that is able to induce neuromodulation between different cells. In that context using the term “rewire/rewiring” is already problematic. The same accounts for using the term “synapse” or “peptidergic synapse” in this context. A synapse is always defined spatially. Although peptidergic synapses exist, the authors do establish de novo peptidergic transmission and neuromodulation but no formation of any de novo synapses. The authors should thoroughly revise the use of the terms within their manuscript.”

Although the term “synapse” was coined by Charles Sherrington sixty years before George Palade showed the existence of junctional synapses, and it was coined to refer to functional connections between excitable cells, we understand the reviewer’s concern that it might be confusing for some readers that might associate “synapse” primarily with junctional connections. To address this we have edited “synapse” to “connections”, a term used in connectomic papers examining neuropeptide connectomes (see <https://pubmed.ncbi.nlm.nih.gov/27984591/>). We also define “rewire” in the manuscript as “the creation or modification of synaptic relationships”, and to emphasize this point, we have edited the manuscript to now read “functional rewiring”. These modifications occur at lines 1, 5, 35, 54, 58, 70, 72, 147, 150, 224, 241, 251, 252, 259, 310, 535, 543, 667) as well as in the title of the manuscript.

R2.3: “The use of the terms presynaptic and postsynaptic for Neuro2a cells in culture should be done consequently using quotation marks as already done at some parts of the text (this also includes the figure legend). Alternatively, cells could be categorized as “sender” and “receiver” depending on which construct of the HySyn system they are expressing.”

We have added the quotation marks as suggested (lines 100, 101, 103, 104, 107, 110, 111, 112, 113, 116, 117, 123, 125, 129, 131, 136, 143, 145, 154, 155, 159, 462, 463, 471, 474, 485, 486, 499, 503, 592, 593, 598, 606, 622, 625).

R2.4: “Line 110: I suggest to not use the term neuron for Neuro2a cells.”

We have integrated this edit as suggested (lines 110 and 276).

R2.5: “Fig.1 b,c,f: It is recommended to use perceptually uniform colormaps instead of “rainbow” colormaps, especially to make the heatmaps barrier free to visually impaired readers. Useful colormaps can be found here for instance: <https://hauselin.github.io/colorpalettejs/>”

We have updated Figure 2 with perceptually uniform colormaps as suggested.

R2.6: “Line 272: It’s “Jena” not “Jenna”.”

We have corrected this in the text (line 373), thank you.

Reviewer #3:

I think the authors have done a good job responding to reviewer comments. I particularly think it is noteworthy that they have provided a lot of additional experimental data which would not have been easy during the pandemic. I am in favor of publication.

Thank you